# Fence off Anomaly Interference: Cross-Domain Distillation for Fully Unsupervised Anomaly Detection

## Abstract

Fully Unsupervised Anomaly Detection (FUAD) is a practical extension of Unsupervised Anomaly Detection (UAD), aiming to detect anomalies without any labels even when the training set may contain anomalous samples. To achieve FUAD, we pioneer the introduction of Knowledge Distillation (KD) paradigm based on teacher–student framework into the FUAD setting. However, due to the presence of anomalies in the training data, traditional KD methods risk enabling the student to learn the teacher's representation of anomalies under FUAD setting, thereby resulting in poor anomaly detection performance. To address this issue, we propose a novel Cross-Domain Distillation (CDD) framework based on the widely studied reverse distillation (RD) paradigm. Specifically, we design a Domain-Specific Training, which divides the training set into multiple domains with lower anomaly ratios and train a domain-specific student for each. Cross-Domain Knowledge Aggregation is then performed, where pseudo-normal features generated by domain-specific students collaboratively guide a global student to learn generalized normal representations across all samples. Experimental results on noisy versions of the MVTec AD and VisA datasets demonstrate that our method achieves significant performance improvements over the baseline, validating its effectiveness under FUAD setting.

## 1 Introduction

In the field of industrial image anomaly detection, acquiring or predefining anomalous samples is often impractical. Consequently, Unsupervised Anomaly Detection (UAD), which relies only on normal samples for training, has been extensively studied [5, 7]. To tackle the challenges of UAD task, a variety of methods are proposed, such as those based on memory banks [26, 2], anomaly synthesis [19, 35], and image reconstruction [4, 1]. In recent years, UAD methods based on Knowledge Distillation (KD) have gained increasing attention [6]. Compared to other techniques, they show greater potential in pixel-level anomaly localization. The KD-based UAD methods typically employ a teacher-student framwork, which allows the student network to imitate the feature representations of the teacher on normal samples. Since the student is never exposed to anomalous samples during training, its ability to generate teacher's anomaly features is limited. This discrepancy in feature mimicking performance becomes a useful signal for anomaly identification.

In real-world scenarios, however, it is often inevitable that the collected data contain a small proportion of anomalous samples. Relying entirely on manual data cleaning incurs high labor costs. This motivates the need for Fully Unsupervised Anomaly Detection (FUAD), a more practical and challenging setting where the training set may contain unlabeled anomalous samples. Although several studies have begun to explore this task, most existing methods rely heavily on memory

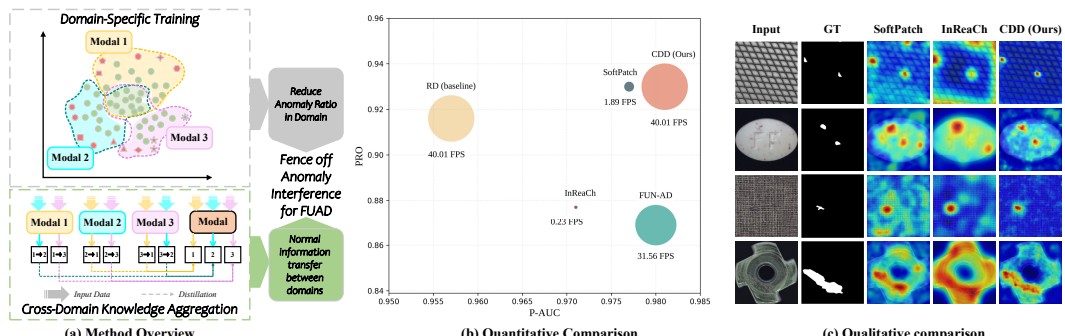

(a) Method Overview    (b) Quantitative Comparison    (c) Qualitative comparison

Figure 1: (a) Simplified schematic of Cross-Domain Distillation with 3 domains, where the top represents Domain-Specific Training and the bottom depicts ross-Domain Knowledge Aggregation. (b) Quantitative comparison against other FUAD methods on MVTec AD-noise-0.1 (with 10% anomaly ratio in the train set) [5], with P-AUC (x-axis), PRO (y-axis), and circle size indicating FPS (larger means faster inference), proves that CDD has the best overall performance. (c) Qualitative comparison with other FUAD methods on MVTec AD-noise-0.1.

banks [17, 24, 16], which introduces additional storage overhead. In contrast, Knowledge Distillation paradigm offers a storage-efficient alternative, yet its potential in FUAD has not been fully explored.

We make two key observations regarding the data under the FUAD setting: (1) From a probabilistic perspective, normal pixels still dominate the training set despite the presence of noise; (2) In terms of feature distribution, teacher representations of normal samples tend to be more compact and stable, making them easier for the student network to learn, whereas anomaly features are more dispersed and less likely to be captured. These insights suggest that, even under the FUAD setting, the student network trained via KD inherently focuses on learning normal representations, leading to poor fitting in anomalous regions. This makes the discrepancy between teacher and student features a reliable signal for anomaly detection, which indicates the feasibility and potential of applying the KD paradigm to the FUAD setting.

However, meanwhile, Knowledge Distillation faces a long-standing over-generalization problem [28, 32, 27]. Even though the student network is trained to learn teacher's representations only on normal pixels, its learned representation ability may still generalize to anomalous ones, which leads to miss detections when testing on anomalous samples. This issue becomes more pronounced under the FUAD setting. If a certain type of anomaly appears frequently in the training data, the student may learn its common feature patterns and become capable of generating teacher-like representations for similar anomalies during inference.

To address the above challenge, we propose a novel cross-domain distillation framework for FUAD, built upon the widely studied KD-based UAD method Reverse Distillation (RD) [9] with an encoder-decoder architecture. First, our intuition is that *reducing the probability of anomalous samples being learned during training mitigates the student's tendency to overfit to techer's anomaly features*. To achieve this, we design a domain division mechanism that distributes high-confidence normal samples across multiple domains while dispersing potentially anomalous samples, thereby lowering the anomaly ratio within each domain without discarding any data. Considering that data distributions vary across domains and that RD's student decoder generates anomaly-free features for unseen anomalous samples, we hypothesize that *domain-specific students trained on different domains produce pseudo-normal features when applied to other domains*. Building on this insight, we introduce a **C**ross-**D**omain **D**istillation (CDD) framework: for each domain, we utilize domain-specific students from other domains to generate pseudo-normal features for its samples, guiding the training of a global student decoder. This global student learns to produce anomaly-free features across all samples, both normal and anomalous. Finally, the distance between the features generated by the global student decoder and the teacher encoder is used to detect and localize anomalies. Our contributions are summarized as follows:

- We are the first to explore the application of the knowledge distillation paradigm to the Fully Unsupervised Anomaly Detection task.
- We propose Domain-Specific Training (DST) as in Fig. 1 (a), which first performs Confidence-Guided Domain Construction to build data domains with low anomaly proba-

bility. Then, each domain is used to train a domain-specific student via Domain-Specific Distillation with Regularization.

- We introduce Cross-Domain Knowledge Aggregation (CDKA), where domain-specific students provide pseudo-normal features for each sample to train a global student that integrates information across all domains as depicted in Fig. 1 (a).

- Experimental verification shows that our CDD is significantly higher than the baseline RD, and has better performance and faster inference speed than the previous FUAD methods.

## 2 Related Work

**Unsupervised Anomaly Detection.** Unsupervised Anomaly Detection (UAD) has been widely studied in recent years due to its ability to operate without requiring anomalous samples during training. Existing methods are broadly categorized into the following types: (1) reconstruction-based generative models [4, 1, 29, 25, 33, 37], which learn to reconstruct only normal samples and identify anomalies based on reconstruction errors during inference; (2) density estimation-based methods [8, 12, 39], which assume that normal samples follow a specific distribution in the feature space and detect deviations from this distribution; (3) synthetic anomaly-based approaches [19, 35, 21, 38], which generate pseudo-anomalies using image transformations, external generators, or diffusion models to enhance the model's ability to perceive anomalies; and (4) methods that incorporate pre-trained models and memory bank mechanisms [26, 2, 15], comparing the features of test samples with those of normal samples to identify anomalies. In recent years, Knowledge Distillation-based UAD methods [6, 20, 28, 32, 27, 3, 22] using the teacher-student framework have emerged as excellent methods for anomaly localization. These methods learn representations of normal regions and detect anomalies by measuring the discrepancy in features between the teacher and student networks on anomalous regions. To mitigate the student's over-generalization to anomalies, some studies introduce heterogeneous architectures or reverse information flow, such as Reverse Distillation [9] and its variants [30, 13, 11, 18, 14, 36], which further improve anomaly detection accuracy.

**Fully Unsupervised Anomaly Detection.** Fully Unsupervised Anomaly Detection (FUAD) has attracted increasing attention, owing to its ability to operate without manual annotations and its suitability for tackling noisy training data in real-world scenarios [31]. Existing methods are categorized as follows: (1) SoftPatch [17], based on PatchCore [26], adopts a memory-based patch-level denoising strategy using noise discriminators to mitigate overconfidence. (2) InReaCh [24] builds detection models by associating high-confidence patch channels across training images. (3) FUN-AD [16] leverages nearest-neighbor distances and class homogeneity, employing an iteratively reconstructed memory bank (IRMB) to handle noisy data. However, these methods often rely on explicit memory banks, which impose storage burdens in practice. Knowledge Distillation has shown strong potential in unsupervised anomaly localization without additional storage, but its application to FUAD remains unexplored. This work aims to explore this promising direction.

## 3 Motivation and Assumptions

### 3.1 Rethinking Reverse Distillation for FUAD

**What is Reverse Distillation?** Early KD-based AD methods typically adopt a homogeneous teacher-student framework, where the student only learns the teacher's representation ability on normal samples. During inference, anomalies are detected by measuring the discrepancy between teacher and student features. Reverse Distillation (RD) [9] builds upon KD by introducing an encoder-decoder structure. The teacher network is a frozen encoder, while the student consists of a trainable one-class bottleneck embedding (OCBE) module $\mathcal{B}(\cdot; \phi)$ and a trainable decoder $\mathcal{D}_S(\cdot; \psi)$.

Let the training set be $\mathcal{I}_{train}$. Given a training image $I_i^{train} \in \mathcal{I}_{train}$, the teacher extracts multi-layer features $\mathcal{F}_{\mathcal{T},i} = \mathcal{T}(I_i^{train}) = \{f_{\mathcal{T},i}^l\}_{l=1}^L$, which are then reconstructed by the student network as $\mathcal{F}_{\mathcal{S},i} = \mathcal{S}(\mathcal{F}_{\mathcal{T},i}; \theta_{\mathcal{S}}) = \{f_{\mathcal{S},i}^l\}_{l=1}^L$. The student network is denoted as $\mathcal{S}(\cdot; \theta_{\mathcal{S}})$, with parameters $\theta_{\mathcal{S}} = \{\phi, \psi\}$. The training objective is to minimize the cosine distance between teacher and student features across all $L = 3$ layers on normal samples as:

$$\cos(f_1, f_2) = \frac{f_1 \cdot f_2}{\|f_1\| \|f_2\|} \tag{1}$$

$$\ell_{cos}(\mathcal{F}_{\mathcal{T}}, \mathcal{F}_{\mathcal{S}}) = \sum_{l=1}^{L} \left( 1 - \cos(f_{\mathcal{T}}^l, f_{\mathcal{S}}^l) \right) \tag{2}$$

$$\arg\min_{\theta_{\mathcal{S}}} \mathbb{E}_{I_i \sim \mathcal{I}_{train}} \ell_{cos}(\mathcal{F}_{\mathcal{T},i}, \mathcal{S}(\mathcal{F}_{\mathcal{T},i}; \theta_{\mathcal{S}})) \tag{3}$$

**Why does RD Work for FUAD?** Although Reverse Distillation (RD) is initially designed for training with only normal samples, it demonstrates strong adaptability in Fully Unsupervised Anomaly Detection. We attribute this to two key factors:

(1) **Probability Perspective** - Dominance of Normal Samples
In industrial scenarios, normal samples are much more common than anomalies, which results in low proportion of anomalous images in the training set. Moreover, anomalies typically occupy only a small region within an image. Consequently, the student network, driven by the dominance of normal samples, primarily learns to represent normal features, while the sparsity of anomalies limits their impact on the optimization process.
(2) **Distribution Perspective** - Concentrated Normal vs. Diverse Anomalous
Normal samples exhibit compact and consistent feature patterns, while anomalous samples are diverse and scattered. This makes it difficult for the student to generalize learned anomalous features.

**Challenges of Applying RD to FUAD.** In FUAD task, the training set naturally includes a certain proportion of anomalous samples. If specific anomaly patterns appear repeatedly during training, the student can easily learn to reconstruct the teacher features of these conmmon anomalies. This results in poor discrimination against similar anomalies during testing and further intensifies over-generalization. Therefore, the key challenge in applying RD to FUAD is *how to prevent the student from modeling common anomaly patterns during training, to ensure that it generates anomaly-free features.*

## 3.2 Assumptions

To address over-generalization problem in FUAD, we propose two assumptions based on the diversity and sparsity of anomalies, guiding the following design of our method Cross-Domain Distillation.

**Assumption 1 (Limited Representation of Rare Anomalies)** *When a particular anomaly type is sufficiently rare in training data, the student fails to learn its corresponding teacher anomaly features, and instead tends to produce features that closely resemble normal patterns.*

Due to the consistency of normal samples and the diversity of anomalies (i.e., anomalies exhibit multiple distinct patterns), we assume the training set contains one normal type and $M_{train}$ anomaly types, expressed as:

$$\mathcal{I}_{\text{train}} = \mathcal{N} \cup \mathcal{A} = \mathcal{N} \cup \bigcup_{m=1}^{M_{train}} \mathcal{A}_m \tag{4}$$

where $\mathcal{N}$ denotes the set of normal samples, $\mathcal{A}_m$ denotes the set of the $m$-th anomaly type, and:

$$\mathbb{P}(\mathcal{N}) \gg \mathbb{P}(\mathcal{A}_m) \quad \forall m = 1, \dots, M_{train} \tag{5}$$

Following Empirical Risk Minimization (ERM), the training objective is to minimize the distance between student features and teacher features over all samples. The empirical risk can be expressed as

$$\mathcal{L} = \mathbb{P}(\mathcal{N}) \cdot \mathbb{E}_{I_i \sim \mathcal{N}}[\ell_{cos}(\mathcal{F}_{T,i}, \mathcal{F}_{S,i})] + \sum_{m=1}^{M_{train}} \mathbb{P}(\mathcal{A}_m) \cdot \mathbb{E}_{I_j \sim \mathcal{A}_m}[\ell_{cos}(\mathcal{F}_{T,i}, \mathcal{F}_{S,i})] \tag{6}$$

The gradient of parameters $\theta_{\mathcal{S}}$ is:

$$\frac{\partial \mathcal{L}}{\partial \theta_{\mathcal{S}}} = \mathbb{P}(\mathcal{N}) \cdot \mathbb{E}_{I_i \sim \mathcal{N}} \left[ \frac{\partial \ell_{cos}}{\partial \theta_{\mathcal{S}}} \right] + \sum_{m=1}^{M_{train}} \mathbb{P}(\mathcal{A}_m) \cdot \mathbb{E}_{I_j \sim \mathcal{A}_m} \left[ \frac{\partial \ell_{cos}}{\partial \theta_{\mathcal{S}}} \right] \tag{7}$$

If $\mathbb{P}(\mathcal{A}_m)$ is small enough, the contribution of the anomaly type $\mathcal{A}_m$ to the gradient is negligible. Thus, the student receives limited learning signals for this type of anomaly and fails to reconstruct the corresponding teacher features effectively.

**Assumption 2 (Lack of Cross-Anomaly Generalization)** *Even if a student learns to reconstruct some specific anomaly patterns during training, this reconstruction ability is not generalized to other unseen anomaly types.*

This assumption is based on the diversity of anomalies. Anomalies are often unstructured and come from different sources or physical mechanisms. As a result, they follow multiple, structurally different patterns in the feature space:

$$\mathcal{F}_{T,i} \mid I_i \in \mathcal{A}_m \sim \mathcal{P}_m \tag{8}$$

where each pattern $\mathcal{P}_m$ represents the teacher's feature distribution for the $m$-th type of anomaly. The total number of types is $M$, which may even be infinite in practice.

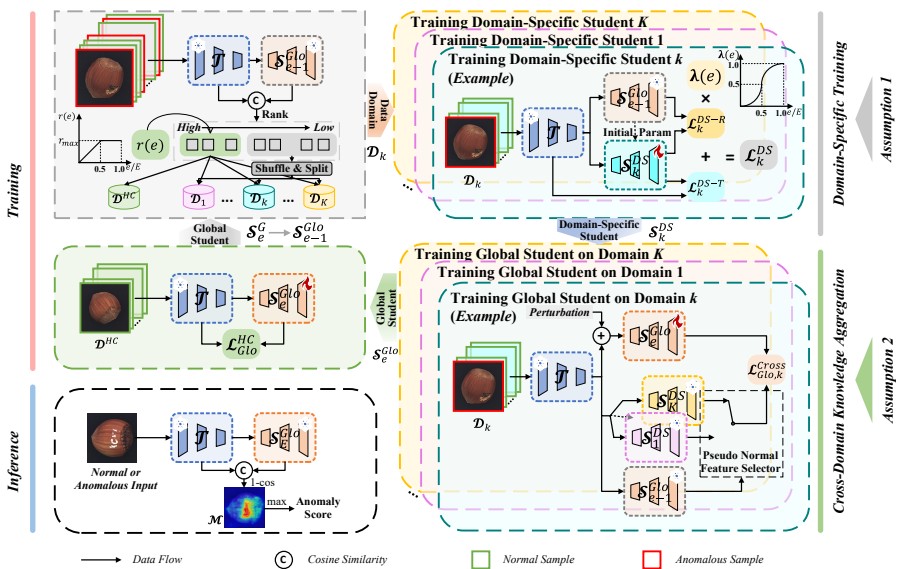

Figure 2: Overall framework of our proposed CDD.

During training, the student network only sees a subset of these anomaly patterns:

$$\mathcal{P}_{\text{train}} = \{\mathcal{P}_1, \ldots, \mathcal{P}_{M_{\text{train}}}\}, \quad M_{\text{train}} \ll M \tag{9}$$

According to the **No Free Lunch** theorem, if an input anomalous sample $I_j \sim \mathcal{P}_{m'}$ with $\mathcal{P}_{m'} \notin \mathcal{P}_{\text{train}}$, its distribution is outside the training support. Then, the student may fail to generate the correct teacher features:

$$\mathcal{F}_{S,j} \not\approx \mathcal{F}_{T,j}, \quad I_j \notin \mathcal{P}_{\text{train}} \tag{10}$$

Since normal samples dominate the training set, the student tends to generate features similar to the normal distribution.

## 4 Method

**Problem Definition.** In FUAD, we denote the training set as $\mathcal{I}_{train} = \{I_i^{train}\}_{i=1}^{N}$, where each image $I_i^{train} \in \{\mathcal{N}, \mathcal{A}\}$ is unlabeled and may be normal or anomalous. The test set $\mathcal{I}_{test} = \{I_j^{test}\}_{j=1}^{M}$ comprises both normal and anomalous images, with normal samples following the same distribution as $\mathcal{I}_{train}$. The objective is to learn the distribution of normal samples from $\mathcal{I}_{train}$ to detect anomalies in $\mathcal{I}_{test}$.

**Overview.** Fig. 2 illustrates the training process of each epoch (top and lower right) and the inference process (lower left). All teacher and student networks follow the design of Reverse Distillation. The teacher is a WideResNet-50 [34] pre-trained on ImageNet [10]. And each student includes an OCBE module and a decoder.

Each training epoch consists of two stages: Domain-Specific Training and Cross-Domain Knowledge Aggregation. In the first stage, we propose Confidence-Guided Domain Construction to extract high-confidence normal samples from the original training set and use them as the intersection between multiple data domains. In this way, each domain has a reduced anomaly ratio compared to the full dataset. Then, we train a domain-specific student for each domain using Domain-Specific Distillation with Regularization. Based on *Assumption 1*, these students ease off from modeling anomaly features and thus focus on modeling normal features in their local domains. The second stage Cross-Domain Knowledge Aggregation mainly explains how to use the domain-specific students obtained in the first stage to train a global student that reconstructs normal features on all samples. According to *Assumption 2*, for anomalous samples in a specific domain $k$, domain-specific students that are not trained on domain $k$ still generates normal-like features. We use these features as pseudo-normal supervision signals to perform Cross-Domain Pseudo-Normal Feature Distillation for the global student. After that, we further distill the global student using the teacher on high-confidence normal samples, enabling it to effectively learn the reliable reconstruction of normal patterns.

The lower left part of Fig. 2 depicts the inference process. During inference, for each image $I_j^{test} \in \mathcal{I}_{test}$, cosine distances across multi-layer features generated by the teacher $\mathcal{T}$ and the global student trained for $E$ epochs $\mathcal{S}_E^{Glo}$ are fused to generate a pixel-level anomaly map $\mathcal{M}$, whose maximum value serves as the image-level anomaly score $s$:

$$\mathcal{M}(h, w) = \sum_{l=1}^{L} \left(1 - \cos(f_{\mathcal{T}}^l(h, w), f_{\mathcal{S}_E^{Glo}}^l(h, w))\right), s = \max(\mathcal{M}) \tag{11}$$

## 4.1 Domain-Specific Training

**Confidence-Guided Domain Construction** Based on *Assumption 1*, reducing the anomaly probability in the training set helps the student better learn normal patterns. A naive way to achieve this is to retain only high-confidence normal samples or discard low-confidence anomalous ones. However, such strategies fail to fully utilize the training data, as potentially useful normal regions are also discarded along with the anomalies.

To address this issue, we introduce a confidence-guided strategy on top of naive equal partitioning. Specifically, we inject a portion of highly confident normal samples into each domain based on normality confidence scores, which ensures that: (1) The anomaly ratio in each domain becomes lower than that in the original training set, reducing the interference of anomalous samples on the modeling of normal patterns. (2) The normal distribution in each domain remains more similar to the overall normal distribution, mitigating the negative impact of domain partitioning on student normality modeling.

We use the features output by the global student of the previous epoch $\mathcal{S}_{e-1}^{Glo}$ as the basis for confidence evaluation. For each training sample $I_i$, the average cosine similarity between teacher features $f_{\mathcal{T}}$ and global student features $f_{\mathcal{S}_{e-1}^G}$ across $L$ layers is calculated to obtain the corresponding $\text{Conf}_i$:

$$\text{Conf}_i = \sum_{l=1}^{L} \left\{ \frac{1}{H_l W_l} \sum_{h=1}^{H_l} \sum_{w=1}^{W_l} \cos(f_{\mathcal{T},i}^l(h,w), f_{\mathcal{S}_{e-1}^{Glo},i}^l(h,w)) \right\} \tag{12}$$

All samples are sorted by confidence in descending order. The top $r(e)$ samples form the high-confidence set $\mathcal{D}^{HC}$. The confidence threshold $r(e)$ increases with training, up to 50%. Let $e$ be the current epoch and $E$ the total epochs, $r(e)$ is calculated as

$$r(e) = \min\left(\frac{e}{E}, 0.5\right) \tag{13}$$

The remaining low-confidence samples are randomly and evenly divided into $K$ subsets, denoted $\mathcal{D}_k^{LC}, k = 1, \ldots, K$. By combining $\mathcal{D}^{HC}$ and $\mathcal{D}_k^{LC}$, each domain is expressed as

$$\mathcal{D}_k = \mathcal{D}^{HC} \cup \mathcal{D}_k^{LC}, \quad k = 1, \ldots, K. \tag{14}$$

**Domain-Specific Distillation with Regularization** After domain construction, we train a corresponding domain-specific student $\mathcal{S}_k^{DS}$ for the $k$-th domain, who learns to reconstruct the features of samples within its corresponding domain. The initial parameters of each domain-specific student are inherited from the global student of the previous epoch $\mathcal{S}_{e-1}^{Glo}$. This training process of $\mathcal{S}_k^{DS}$ follows the basic framework of Reverse Distillation, which minimizes the cosine distance between the features generated by the student $\mathcal{F}_{\mathcal{S}_k^{DS}}$ and the features of the teacher $\mathcal{F}_{\mathcal{T}}$. In this way, the domain-specific students are able to model the teacher's feature representation ability of data in their local domain.

However, even with controlled anomaly ratios and dispersed common anomalies, the domain-specific student may still learn representations of abnormal samples, especially when a particular type of anomaly is overly represented in the domain. To further tackle this problem, we introduce pseudo-normal features generated by the global student obtained from the previous epoch $\mathcal{S}_{e-1}^{Glo}$ as the regularization signal. As the global student becomes more and more capable of modeling normal patterns during training, it provides useful guidance to help domain-specific students avoid over-learning anomaly features. The loss $\mathcal{L}_k^{DS}$ used to train each domain-specific student $\mathcal{S}_k^{DS}$ combines two terms: the primary distillation loss (from the teacher) and the regularization loss (from the global student), which is expressed as

$$\mathcal{L}_k^{DS} = \mathbb{E}_{I_i \sim \mathcal{D}_k} (\underbrace{\ell_{cos}(\mathcal{F}_{\mathcal{T},i}, \mathcal{F}_{\mathcal{S}_k^{DS},i})}_{\text{Teacher Guidance } \mathcal{L}_k^{DS-T}} + \lambda(e) \cdot \underbrace{\ell_{cos}(\mathcal{F}_{\mathcal{S}_{e-1}^{Glo},i}, \mathcal{F}_{\mathcal{S}_k^{DS},i})}_{\text{Regularization } \mathcal{L}_k^{DS-R}}) \tag{15}$$

where $\lambda(e)$ is a dynamic increasing coefficient that adjusts the regularization strength over the training epochs. It is controlled using an S-shaped scheduling function with $p = 4.0$ as

$$\lambda(e) = \frac{(e/E)^p}{(e/E)^p + (1 - e/E)^p} \tag{16}$$

## 4.2 Cross-Domain Knowledge Aggregation

**Cross-Domain Pseudo-Normal Feature Distillation** Due to the high consistency of normal samples across domains, domain-specific students reconstruct correct normal features on normal samples in all domains. Based on *Assumption 2*, the diversity of anomalies prevents domain-specific students from generalizing to out-of-domain anomaly patterns, even if they learn the reconstruction of anomaly features in local domains. Following this idea, we propose using domain-specific students to generate pseudo-normal features for out-of-domain samples. providing supervision for the training of the global student to generate normal features on all samples. To prevent pseudo-normal feature contamination caused by some domain-specific students learning the ability to

Table 1: Anomaly detection and localization results I-AUC / P-AUC / PRO under *No Overlap* setting on MVTec AD-noise-0.1 with the best in bold. and the second best underlined.

| Category | Unsupervised | | Fully Unsupervised | | | |
|---|---|---|---|---|---|---|
| | RD [9] | URD [23] | SoftPatch [17] | InReaCh [24] | FUN-AD [16] | CDD (Ours) |
| bottle | 0.997 / 0.983 / 0.955 | 0.992 / 0.984 / 0.961 | 1.000 / 0.986 / 0.956 | 1.000 / 0.981 / 0.915 | 1.000 / 0.992 / 0.960 | 1.000 / 0.987 / 0.959 |
| cable | 0.931 / 0.835 / 0.768 | 0.955 / 0.881 / 0.824 | 0.996 / 0.984 / 0.919 | 0.958 / 0.978 / 0.862 | 0.952 / 0.920 / 0.740 | 0.981 / 0.969 / 0.891 |
| capsule | 0.939 / 0.980 / 0.956 | 0.951 / 0.981 / 0.958 | 0.961 / 0.990 / 0.965 | 0.446 / 0.914 / 0.657 | 0.922 / 0.987 / 0.855 | 0.942 / 0.984 / 0.950 |
| carpet | 0.985 / 0.988 / 0.957 | 0.993 / 0.991 / 0.972 | 0.989 / 0.992 / 0.959 | 0.980 / 0.992 / 0.958 | 1.000 / 0.995 / 0.953 | 0.989 / 0.989 / 0.960 |
| grid | 0.956 / 0.994 / 0.979 | 1.000 / 0.990 / 0.976 | 0.965 / 0.991 / 0.963 | 0.917 / 0.983 / 0.929 | 0.991 / 0.993 / 0.935 | 1.000 / 0.992 / 0.976 |
| hazelnut | 1.000 / 0.992 / 0.936 | 0.994 / 0.993 / 0.953 | 1.000 / 0.994 / 0.942 | 1.000 / 0.988 / 0.907 | 0.999 / 0.991 / 0.885 | 1.000 / 0.993 / 0.945 |
| leather | 1.000 / 0.995 / 0.988 | 1.000 / 0.995 / 0.990 | 1.000 / 0.994 / 0.988 | 1.000 / 0.992 / 0.985 | 1.000 /0.998 / 0.986 | 1.000 / 0.991 / 0.971 |
| metal_nut | 0.988 / 0.833 / 0.859 | 0.994 / 0.848 / 0.869 | 0.998 / 0.886 / 0.838 | 0.970 / 0.958 / 0.887 | 0.997 / 0.992 / 0.864 | 1.000 / 0.962 / 0.870 |
| pill | 0.960 / 0.966 / 0.956 | 0.961 / 0.956 / 0.950 | 0.953 / 0.977 / 0.945 | 0.889 / 0.956 / 0.883 | 0.939 / 0.972 / 0.893 | 0.971 / 0.978 / 0.958 |
| screw | 0.980 / 0.995 / 0.983 | 0.954 / 0.994 / 0.977 | 0.952 / 0.994 / 0.975 | 0.779 / 0.982 / 0.936 | 0.913 / 0.981 / 0.772 | 0.934 / 0.992 / 0.974 |
| tile | 0.988 / 0.961 / 0.858 | 1.000 / 0.964 / 0.897 | 1.000 / 0.959 / 0.878 | 0.999 / 0.965 / 0.878 | 0.999 / 0.978 / 0.939 | 0.997 / 0.955 / 0.879 |
| toothbrush | 1.000 / 0.991 / 0.939 | 1.000 / 0.992 / 0.943 | 1.000 / 0.986 / 0.915 | 0.990 / 0.989 / 0.904 | 0.972 / 0.981 / 0.850 | 0.997 / 0.987 / 0.916 |
| transistor | 0.943 / 0.882 / 0.753 | 0.948 / 0.901 / 0.812 | 0.996 / 0.952 / 0.819 | 0.929 / 0.982 / 0.786 | 0.962 / 0.975 / 0.520 | 0.998 / 0.980 / 0.831 |
| wood | 0.990 / 0.978 / 0.906 | 0.994 / 0.983 / 0.924 | 0.997 / 0.979 / 0.912 | 0.947 / 0.962 / 0.875 | 1.000 / 0.977 / 0.960 | 0.993 / 0.979 / 0.916 |
| zipper | 0.924 / 0.976 / 0.941 | 0.861 / 0.973 / 0.926 | 0.974 / 0.989 / 0.969 | 0.952 / 0.937 / 0.796 | 0.984 / 0.970 / 0.925 | 0.958 / 0.980 / 0.950 |
| Average | 0.972 / 0.957 / 0.916 | 0.973 / 0.962 / 0.929 | **0.985** / 0.977 / **0.930** | 0.917 / 0.971 / 0.877 | 0.975 / 0.980 / 0.869 | 0.984 / **0.981** / **0.930** |

reconstruct certain types of teacher anomaly features, we design a Consensus-driven Pseudo-Normal Feature Selection strategy.

Specifically, we select the most "consensual" domain-specific student to generate the normal feature supervision for each sample. The core motivation is that for the same sample, multiple domain-specific students that have not been trained on the sample should generate similar normal features. In the implementation, we achieve pseudo-normal feature selection by eliminating outlier features that are more likely to be abnormal features from the output features of domain-specific students with the help of the global student from the previous epoch $\mathcal{S}_{e-1}^{Glo}$.

For a sample $I_i$ from domain $\mathcal{D}_k$, we first extract features $\mathcal{F}_{\mathcal{S}_h^{DS},i} = \{f_{\mathcal{S}_h^{DS},i}^l\}_{l=1}^L, h = \{1, \ldots, K\} \setminus k$ using the domain-specific students from domains $\mathcal{D}_h, h = \{1, \ldots, K\} \setminus k$, and obtain the reference features $\mathcal{F}_{\mathcal{S}_{e-1}^{Glo},i} = \{f_{\mathcal{S}_{e-1}^{Glo},i}^l\}_{l=1}^L$ the global student from the previous epoch. We construct an affinity matrix $\text{Aff}_i \in \mathbb{R}^{(K-1)\times 1}$, where each element measures the cosine similarity between the flattened features of each student and the global student:

$$\text{Aff}_i(h) = \sum_{l=1}^L \cos(f_{\mathcal{S}_h^{DS},i}^l, f_{\mathcal{S}_{e-1}^{Glo},i}^l), \quad h = \{1, \ldots, K\} \setminus k \tag{17}$$

The pseudo-normal feature for the training sample is then selected as the one with the highest similarity:

$$\mathcal{F}_{pseudo,i} = \mathcal{F}_{\mathcal{S}_{h^*}^{DS},i}, \quad h^* = \arg\max_h \text{Aff}_i(h) \tag{18}$$

However, the selected pseudo-normal features may still be contaminated with anomaly features. To prevent the trainable global student from overfitting these pseudo-normal features, we inject Gaussian noise with $\sigma_{noise} = 0.2$ as feature perturbation into its input:

$$\mathcal{F}_{\mathcal{S}_e^{Glo},i}^* = \mathcal{S}(\mathcal{F}_{\mathcal{T},i} + \delta; \theta_{\mathcal{S}^{Glo}}), \quad \delta \sim \mathcal{N}(0, \sigma_{noise}^2) \tag{19}$$

The loss of Cross-Domain Pseudo-Normal Feature Distillation $\mathcal{L}_{Glo}^{Cross}$ is then defined as:

$$\mathcal{L}_{Glo}^{Cross} = \sum_{k=1}^K \mathbb{E}_{I_i \sim \mathcal{D}_k} \ell_{cos}(\mathcal{F}_{pseudo,i}, \mathcal{F}_{\mathcal{S}_e^{Glo},i}^*) \tag{20}$$

**Confident Distillation for High-Confidence Domain** In addition to pseudo-normal feature guidance, we also leverage the previously defined high-confidence sample set $\mathcal{D}^{HC}$, using teacher features as direct supervision to further enhance the global student's ability to model true normal patterns:

$$\mathcal{L}_{Glo}^{HC} = \mathbb{E}_{I_i \sim \mathcal{D}^{HC}} \ell_{cos}(\mathcal{F}_{\mathcal{T},i}, \mathcal{F}_{\mathcal{S}_e^{Glo},i}) \tag{21}$$

# 5 Experiments

## 5.1 Experimental Setup

**Datasets.** We conduct experiments on two widely-used datasets: MVTec AD and VisA. Since both datasets are originally designed for unsupervised anomaly detection, we adapt them to the FUAD setting following SoftPatch [17]. Specifically, we keep the normal training images unchanged and randomly inject a portion of anomalous test samples into the training set with a predefined anomaly ratio $R_{\text{noise}}$. We evaluate under two settings: (1) *No overlap* setting, where injected anomalous samples are removed from the test set; and (2) *Overlap* setting, where these anomalies remain in the test set, making the task more challenging.

Table 2: Anomaly detection and localization results I-AUC / P-AUC / PRO under *Overlap* setting on MVTec AD-noise-0.1 with the best in bold. and the second best underlined.

| | Unsupervised | | Fully Unsupervised | | | |
| --- | --- | --- | --- | --- | --- | --- |
| | RD [9] | URD [23] | SoftPatch [17] | InReaCh [24] | FUN-AD [16] | CDD (Ours) |
| Average | 0.708 / 0.818 / 0.901 | 0.696 / 0.792 / 0.909 | **0.984** / 0.957 / 0.915 | 0.879 / 0.943 / 0.861 | 0.976 / **0.977** / 0.870 | 0.971 / 0.973 / **0.921** |

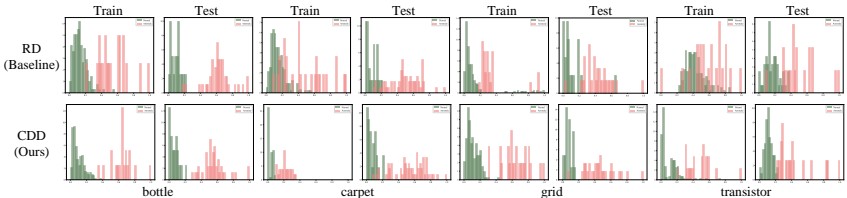

(a) I-AUC      (b) P-AUC      (c) PRO

Figure 3: Comparison of anomaly detection performance with baseline RD under different $R_{\text{noise}}$.

**Implementation Details.** We train a separate model for each category. The backbone follows RD, using a WideResNet-50 pretrained on ImageNet. Following SoftPatch [17], all images are resized to $256 \times 256$ and then center cropped to $224 \times 224$ during both training and inference. All domain-specific students and the global student are optimized by its own Adam optimizer with a learning rate of 0.005 and trained for 200 epochs. To smooth the obtained anomaly maps, we apply Gaussian filtering with $\sigma = 4$. All our experiments are performed on a single Nvidia GTX 3090 GPU.

**Evaluation Metrics.** We use the area under the ROC curve (AUC) at both image and pixel levels, denoted as I-AUC and P-AUC, to evaluate anomaly detection and localization performance. The per-region-overlap (PRO) metric is also reported to better evaluate the localization performance of anomalies with small sizes.

## 5.2 Anomaly Detection under FUAD setting

**Results on MVTec AD.** We evaluate our proposed Cross-Domain Distillation (CDD) on the MVTec-AD dataset with $R_{\text{noise}} = 0.1$, denoted as MVTec-AD-noise-0.1. CDD is compared with unsupervised KD-based UAD methods including RD [9], and FUAD methods, such as SoftPatch [17], InReaCh [24], and FUN-AD [16]. Tab. 1 and Tab. 2 present the anomaly detection and localization results under *No Overlap* and *Overlap* settings, respectively, where each method reports I-AUC, P-AUC, and PRO metrics, all reproduced through 200 epochs of model training under a unified dataset split. In the *No Overlap* setting, CDD matches SoftPatch's I-AUC while achieving a P-AUC of 0.981 and PRO of 0.930, surpassing all methods including RD in pixel-level localization. In the *Overlap* setting, despite some methods' performance dropping sharply, CDD retains robustness with a P-AUC of 0.973 and PRO of 0.921, significantly outperforming the baseline and demonstrating strong resistance to anomaly noise. Furthermore, we compare RD and CDD on MVTec AD-noise-{0.2-0.15} as in Fig. 3. At low $R_{\text{noise}}$, CDD's advantage over RD is subtle, but as $R_{\text{noise}}$ rises, RD becomes unstable, especially in the *Overlap* setting, while CDD shows consistent performance with minimal fluctuations.

**Results on VisA.** For the VisA dataset, we set $R_{\text{noise}} = 0.05$ (VisA-noise-0.05) based on the ratio of normal to anomalous samples in the original dataset and conduct relevant experiments as in Tab. 3. The compared methods include unsupervised and fully unsupervised AD methods. Our method achieves the best performance in both *No Overlap* and *Overlap* settings. Notably, in the *Overlap* setting, we outperform the baseline RD by 28.0% in I-AUC, 6.8% in P-AUC, and 1.9% in PRO, respectively, demonstrating that

Table 3: Anomaly detection and localization results on VisA-noise-0.05.

| Setting | Metrics | RD [9] | SoftPatch [17] | InReaCh [24] | CDD (Ours) |
| --- | --- | --- | --- | --- | --- |
| *No Overlap* | I-AUC | 0.945 | 0.927 | 0.827 | **0.954** |
| | P-AUC | 0.979 | **0.985** | 0.974 | 0.982 |
| | PRO | 0.897 | 0.985 | 0.904 | **0.911** |
| *Overlap* | I-AUC | 0.656 | 0.924 | 0.725 | **0.936** |
| | P-AUC | 0.909 | 0.954 | 0.914 | **0.977** |
| | PRO | 0.892 | 0.883 | 0.721 | **0.911** |

our cross-domain training strategy effectively enhances the baseline's resilience to anomaly interference.

Figure 4: Comparison of histograms of anomaly scores obtained by RD and our CDD.

**Visualization Comparisons.** We perform additional visualization experiments to compare our proposed CDD with the baseline RD. First, we obtain anomaly scores on both the training and test sets of MVTec-AD-noise-0.1 using the trained RD and CDD, generating histograms of anomaly scores for all the samples as depicted in Fig. 4. On one hand, RD proves effective in the FUAD setting, yet it inadvertently learns certain anomaly patterns from the training set, impairing its ability to accurately detect anomalies. Notably, our CDD overcomes this limitation, markedly improving anomaly detection ability on the training set.

Figure 5: Visualization comparison of anomaly maps generated by RD and our CDD.

Fig. 5 further compares anomaly maps generated by RD and CDD. Compared to RD, CDD exhibits greater sensitivity to anomalies, intuitively demonstrating its ability to mitigate overfitting to some extent, preventing the student network from excessively learning the teacher's anomaly representations.

Table 4: Ablation study of module effectiveness on MVTec AD-noise-0.1 with $K = 2$.

| DST | | | CDKA | | | | | | |
|------|---------|------|------|------|--------|-------|-------|--------|---------|
| D.C. | Conf.G. | Reg. | P.N. | F.P. | Conf.D. | I-AUC | P-AUC | PRO | Average |
| - | - | - | - | - | - | 0.9721 | 0.9566 | 0.9156 | 0.9481 |
| - | - | - | ✓ | - | - | 0.9701 | 0.9731 | 0.9225 | 0.9552 |
| ✓ | ✓ | - | ✓ | - | - | 0.9709 | 0.9764 | 0.9223 | 0.9565 |
| ✓ | ✓ | - | ✓ | - | ✓ | 0.9761 | 0.9779 | 0.9230 | 0.9590 |
| ✓ | ✓ | - | ✓ | ✓ | ✓ | 0.9802 | 0.9821 | 0.9287 | 0.9637 |
| ✓ | ✓ | ✓ | ✓ | ✓ | ✓ | 0.9836 | 0.9818 | 0.9287 | 0.9647 |

## 5.3 Ablation Analysis

**Effectiveness of Proposed Designs.** We first conduct the stepwise ablation experiments to evaluate the effectiveness of module designs based on $K = 2$ as in Tab. 4. Without any additional designs, the setup reverts to the baseline RD. For Domain-Specific Training (DST), *D.C.* represents a simple even-split domain construction, while *C.G.* integrates Confidence-Guided Domain Construction for improved division. Initially, domain-specific training relies exclusively on the teacher for supervision, with *Reg.* introducing regularization by distilling from the previous global student. For Cross-Domain Knowledge Aggregation (CDKA), *P.N.* denotes the basic cross-domain distillation, generating pseudo-normal features across domains for feature distillation. The inclusion of *F.P.* involves applying feature perturbation to the global student's input during training. Lastly, *Conf.D.* refers to training the global student directly on High-Confidence Domains to learn teacher representations. The results in Tab. 4 confirm that the addition of each module consistently enhances performance over the baseline.

Table 5: Ablation study of domain number $K$ on MVTec-noise-0.1.

| $K$ | I-AUC | P-AUC | PRO | Average |
|-----|-------|-------|-----|---------|
| 2 | 0.9836 | **0.9818** | 0.9287 | 0.9647 |
| 3 | 0.9821 | 0.9793 | 0.9252 | 0.9622 |
| 4 | 0.9791 | 0.9811 | 0.9271 | 0.9624 |
| {2,3,3,2} | **0.9840** | 0.9812 | **0.9297** | **0.9650** |
| {2,3,4,3,2} | 0.9837 | 0.9806 | 0.9260 | 0.9634 |

Table 6: Ablation study of pseudo-normal feature selection strategies on MVTec-noise-0.1.

| | $K = 3$ | | | | |
|---|----------------|-------|-------|-----|---------|
| | Select Strategy | I-AUC | P-AUC | PRO | Average |
| | *All* | 0.9753 | 0.9747 | 0.9251 | 0.9584 |
| One | *Next* | 0.9510 | 0.9692 | 0.9142 | 0.9448 |
| | *Consensual* | **0.9821** | **0.9793** | **0.9252** | **0.9622** |

**Number of Domains.** To investigate the impact of the number of domains $K$, we conduct an ablation study on MVTec-AD, with performance results in Tab. 5. Moreover, we observe that as training progresses, the student can gradually generate normal teacher features. In this case, appropriately increasing $K$ better isolate anomalies. In the later stages, as the global student learns to generate normal features even in anomaly regions, finer domain division becomes less critical, allowing $K$ to be reduced. To test this, we experimented with dynamic $K$ strategies. Results show that the $\{2, 3, 3, 2\}$ strategy achieves a PRO of 0.9297, a 1% improvement over the fixed $K = 2$. This indicates that dynamically adjusting $K$ effectively balances anomaly suppression and normal feature modeling. Therefore, our final design adopts $K$ varying as $\{2, 3, 3, 2\}$ across epochs.

**Selection of Pseudo-Normal Features.** We conduct an ablation study on pseudo-normal feature selection strategies, all performed with $K = 3$, with results presented in Tab. 6. One strategy, labeled *All*, uses pseudo-normal features generated by domain-specific students from all other domains for distillation. Alternatively, we select features from only one domain, either via our Consensus-driven Pseudo-Normal Feature Selection (denoted as *Consensual*) or by choosing the next domain's feature (denoted as *Next*, akin to random selection). Results show that our *Consensual* strategy markedly achieves the best performance, which demonstrates that the Consensus-driven strategy significantly enhances cross-domain distillation quality.

## 6 Conclusions

In this paper, we propose a novel Cross-Domain Distillation framework to address the FUAD task. To reduce the impact of anomalies during training, we introduce two key strategies: Domain-Specific Training, which constructs multiple low-anomaly domains and trains corresponding domain-specific students; and Cross-Domain Knowledge Aggregation, which transfers pseudo-normal features in a cross-domain manner to guide a global student. Compared with the original Reverse Distillation (RD) baseline, our approach significantly improves robustness and accuracy under noisy training conditions. Compared with the original RD baseline, CDD is less affected by anomaly interference under the FUAD setting, as supported by our experimental results.

**Discussion.** Although CDD is implemented based on the RD paradigm, the core design is conceptually general and could be extended to other UAD paradigms. However, our experiments are restricted to RD-based architectures. Future work will focus on adapting and validating CDD under other paradigms, such as feature reconstruction, to further demonstrate its generality.

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
