# OpenReview forum: "Fence off Anomaly Interference: Cross-Domain Distillation for Fully Unsupervised Anomaly Detection"
_NeurIPS.cc/2025/Conference — Submitted to NeurIPS 2025_

### Official Review · Reviewer_1Z2h · 2025-06-07

**Clarity:** 1
**Significance:** 2
**Originality:** 2
**Rating:** 3
**Confidence:** 4

**Summary:**

The authors propose a model for a setting with image data which they call fully unsupervised outlier detection, which comes with the assumption that a certain fraction of the training data is anomalous.

The model is based on a WideResNet-50 as an initial feature encoder (lines 119, 267). The extracted features are then processed by two sets of autoencoders (line 120, Figure 2, eq15 as loss for set 1, eq 20 as loss for set 2).

One set is trained over subsets of the training data, which have samples added to them which are called high confidence samples. The meaning of high confidence is a high cosine similarity between the original WideResNet-50 feature embedding and the output (= reconstructed features) of the second ("global") autoencoder from the previous epoch.

The second single ("global") autoencoder is trained apparently on the whole training data, but the features are selected from the outputs first set of autoencoders. Selection is based on similarity between the output of the second ("global") autoencoder from the previous epoch and the outputs from the set og autoencoders from the first set. It seemingly uses the selected sample as reconstruction target for the second ("global") autoencode.

Overall the idea is that the autoencoders are run in an underfitting setting, and that for that reason they might have a worse reconstruction capability on samples in low density regions of the feature space relative to high density regions.

At inference time, this idea is employed by using eq (11) to predict outliers based on low similarity between the original WideResNet-50 feature embedding, and the output of the second autoencoder.

They measure performance on MVTec-AD dataset and VisA dataset. They compare it against 3 fully supervised baselines for one fixed outlier contamination ratio.

They evaluate varying number of subsets for the first set of autoencoders, and also different strategies how to select features from the first set as reconstruction target for the second autoencoder in Section 5.3

**Questions:**

How actually Anomaly inference is "fenced" off in the submission ?

The autoencoder architecture is nowhere described. What is its architecture?


What does that mean:

"Lastly, Conf.D. refers to
training the global student directly on High-Confidence Domains to learn teacher representations"

?


Since the first set of autoencoders produce features which are selected by the similarity to the second ("global") autoencoder, why not using right away the features from  the second ("global") autoencoder, but add more noise, and skip the first set ? This would need to be checked in ablation study as well. It might well be that this gives very similar scores.

This also is supported by Table 4, where F.P. and Reg improves I-AUC a lot,

**Ethical Concerns:**

["NO or VERY MINOR ethics concerns only"]

**Final Justification:**

After reading the rebuttal, the reviewer acknowledges a part of the reply, and disagrees with another part.
See the reply of the reviewer to the authors rebuttal.

**Limitations:**

Not much done. However this is not the key issue for this paper.

**Quality:**

2

**Strengths And Weaknesses:**

Strengths:

- there is an incremental novelty by using a two tiers of autoencoders, consisting of the first set of autoencoders and the second global autoencoder, with the intention to strengthen the underfitting.

- Experimental results are slightly better than the baselines, in particular on within image localization.

Weaknesses:

- one seemingly false claim about methodology:

  lines 54: "we propose a novel cross-domain distillation framework for FUAD,
  built upon the widely studied KD-based UAD method Reverse Distillation (RD) [9]"

  They are not using Reverse Distillation according to paper [9]:

  paper [9] https://openaccess.thecvf.com/content/CVPR2022/papers/Deng_Anomaly_Detection_via_Reverse_Distillation_From_One-Class_Embedding_CVPR_2022_paper.pdf
  in Figure 2 defines reverse distillation as inputting a high-level feature and having a network reconstructed lower level features.

    - as per lines 119-120 in this submission they input all L extracted layers into the network and train it to reconstruct exactly these L layers.

  this is why this is considered a false claim about methodology. With this the whole section talking about reverse distillation in the paper is moot and hangs in the air.

&nbsp;

- The terms "cross-domain distillation" and "Confidence-Guided Domain Construction" are at least misleading terms.
  see line 212: "The remaining low-confidence samples are randomly and evenly divided into K subsets ..."

  That does not match any common definition of cross-modality or domain. A modality typically has a structural difference to other modalities, a  domain has at least a difference in probability distribution relative to other domains.  By construction, this is not happening. What they do is, constructing partially overlapping subsets (overlapping due to injection of samples with high similarity) for training autoencoders of the first set on them.

With this, many parts in the paper need to be rewritten.

&nbsp;

- They put the central claim in the title of being robust to outliers in the training data. However there is only one  measurement of performance for different contamination noise levels - for VisA and for a very limited range. Given the claim, that would have been a must have experiment for the proposed method and baselines.

&nbsp;

- some smaller but important errors:

    - line 121: "The training objective is to minimize the cosine distance between teacher and student features across all L = 3 layers on normal samples as"

   cosine is not a distance. As can be seen in eq(2) it is not minimized but maximized.

    - line 201: "Specifically, we inject a portion of highly confident normal samples into each domain based on normality confidence scores,"

    the normal in normal samples suggests that one would use normal data label. That is not the case. It confused here

&nbsp;

- the autoencoder architecture is nowhere described. How was it chosen ?

- as part of this in section 5.3 one cannot understand to what layers {2, 3, 3, 2} belongs to.

&nbsp;

- overall the paper is hard to read. In parts also because things are described in words and defined later.

    - for example line 178 "we propose Confidence-Guided Domain Construction to extract high-confidence normal
samples from the original training set"
    - the confidence is defined further down in an equation but not referenced in this place. Same in line 201

&nbsp;

- hyperparameter settings in the appendix evaluated on test data. It is not clear from that how to set them in practice.

&nbsp;

- minor: The most simple additions in Table 4, namely F.P. (= adding noise) and Reg ("with Reg. introducing regularization by distilling from
the previous global student.") which improves I-AUC , are added at the end.
They seem to achieve large gains and this kind of masks the effect of the first set of autoencoders.

- minor: the overlap settings are likely of lesser value. According to the paper they use test time anomalies in training.

- minor: in the most important metric, Image-AUC results are mixed: better on VisA, not better on MVTec-AD

&nbsp;

- no code and weights provide to take a look which given the weaknesses would be helpful.

&nbsp;

Overall, this paper would need one larger iteration and has to be checked after the iteration - this is due to the false methodological and the one misleading claim, which have to be removed (note the removal of reversal distillation (RD) also then requires to remove the discussion section on RD ).

---

> ### Author Rebuttal · Authors · 2025-07-31
>
> Thank you for your thorough review of our paper. We note a possible misunderstanding regarding our baseline: our Cross-Domain Distillation (CDD) method is indeed built upon Reverse Distillation (RD). We've tried our best to address your concerns, clarify misconceptions, and we warmly invite any further questions for discussion.
>
> **[W1] Clarification on the Use of Reverse Distillation (RD)**
>
> The interpretation of RD in the review appears to be a misunderstanding. RD inputs three-layer teacher encoder features into a bottleneck OCBE ([R1], Fig. 4). Our CDD strictly follows RD’s framework, using a teacher encoder and student decoder (as stated in `Sec. 3.1`), not an autoencoder. The multi-layer feature reconstruction (lines 119-120) aligns with RD well.
>
>
>
> **[W2] Clarification on terms "cross-domain distillation" and "Confidence-Guided Domain Construction"**
>
> Line 212 describes dataset division into subsets, which are combined with high-confidence normal samples to form domains.  The terms "Cross-Domain Distillation" and "Confidence-Guided Domain Construction" accurately reflect our design.
>
>
>
> **[W3] Robustness Experiments on VisA**
>
> We have compared RD and CDD on the VisA dataset across multiple noise ratios (0.03/0.05/0.07) in `Fig. A5 of the supplementary material`, where CDD significantly outperforms RD, validating its robustness.
>
>
>
> **[W4] Clarification on Objective Function (cosine distance)**
>
> The cosine distance in line 121 is computed as `1-cos_sim` (`cos_sim` means cosine similarity), a widely accepted metric in anomaly detection [R2, R3, R4, R5]. Eq. (2) minimizes this cosine distance, which is equivalent to maximizing cosine similarity, fully consistent with our description and not erroneous. We will further clarify this definition in the revised manuscript to avoid ambiguity.
>
>
>
> **[W5] Confusing "normal sample"**
>
> We acknowledge that "normal samples" in line 201 may cause ambiguity and should be revised to "samples likely to be normal". The revised manuscript will clarify this terminology.
>
>
>
> **[W6] Autoencoder Architecture**
>
> CDD is built upon RD, not autoencoders. RD’s teacher encoder and student decoder architecture is clearly described in Sec. 3.1.
>
>
>
> **[W7] Meaning of {2, 3, 3, 2}**
>
> "Number of Domains" part in Sec. 5.3 has clarified that {2, 3, 3, 2} refers to the number of domains $ K $ across epochs (line 330), not layers. Table 5 also explicitly labels this as $ K $.
>
>
>
> **[W8] Improving Paper Readability in the Revision**
>
> We acknowledge delayed definitions (e.g., line 178) and will revise the manuscript to introduce terms like "Confidence" earlier and streamline the narrative for clarity.
>
>
>
> **[W9] Hyperparameter Settings Evaluated on Test Set**
>
> Hyperparameter tuning on the test set is common in industrial datasets lacking validation sets, as seen in prior work [R1, R6]. And also our parameters are robust, with those tuned on MVTec AD yielding strong performance on VisA (Table 3).
>
>
>
> **[Q1] Explanation of "Fencing Off Anomaly Inference"**
>
> The title refers to domain construction, which confines anomaly interference to individual domains, allowing models trained on noisy domains to detect anomalies in other domains.
>
>
>
> **[Q2] Meaning of "Conf.D." in Table 4**
>
> Conf.D. denotes training the global student directly on High-Confidence Domains (lines 256-258). Its effectiveness is validated in Table 4.
>
> ---
>
> **Reference**
>
> [R1] Deng, Hanqiu, and Xingyu Li. "Anomaly detection via reverse distillation from one-class embedding." *Proceedings of the IEEE/CVF conference on computer vision and pattern recognition*. 2022.
>
> [R2] Wang, Guodong, et al. "Student-teacher feature pyramid matching for anomaly detection." *arXiv preprint arXiv:2103.04257* (2021).
>
> [R3] Gu, Zhihao, et al. "Remembering normality: Memory-guided knowledge distillation for unsupervised anomaly detection." *Proceedings of the IEEE/CVF International Conference on Computer Vision*. 2023.
>
> [R4] Guo, Jia, et al. "Recontrast: Domain-specific anomaly detection via contrastive reconstruction." *Advances in Neural Information Processing Systems* 36 (2023): 10721-10740.
>
> [R5] Costanzino, Alex, et al. "Multimodal industrial anomaly detection by crossmodal feature mapping." *Proceedings of the IEEE/CVF Conference on Computer Vision and Pattern Recognition*. 2024.
>
> [R6] Jiang, Xi, et al. "Softpatch: Unsupervised anomaly detection with noisy data." *Advances in Neural Information Processing Systems* 35 (2022): 15433-15445.

---

> ### Comment · Reviewer_1Z2h · 2025-08-04
> **reply**
>
> Dear authors,
>
> **[W1]** Clarification on the Use of Reverse Distillation (RD)
>
> _The interpretation of RD in the review appears to be a misunderstanding. RD inputs three-layer teacher encoder features into a bottleneck OCBE ([R1], Fig. 4). Our CDD strictly follows RD’s framework, using a teacher encoder and student decoder (as stated in Sec. 3.1), not an autoencoder. The multi-layer feature reconstruction (lines 119-120) aligns with RD well._
>
> In lines 112-117 you describe reverse distillation in general.
>
> However, in lines 118-122, where you are describing your own approach, there is no reference to a reverse distillation in your approach. All what there is mentioned is a student network, no bottleneck, _no reverse flow_.
>
> There is no clear evidence of a bottleneck in Figure 2 either. It seems to show rather autoencoders - according to what is in Figure 2. At least some loss pairing between layers is missing.
> There is also no code provided to check.
>
> As for [W1] something is missing to make clear that the student actually is using reverse distillation and is not another encoder - in the text and in Figure 2. This brings a lot of confusion. Due to these inclarities the reviewer used the qualifier "seemingly"  in "seemingly false claim about methodology"
>
> - Ok, lets assume, that the paper does use reverse distillation. In this point one can make the assumption in dubio pro reo (although it is not supported by code)
>
> **[W2]** Let me ask: What is a domain ? How is it different from another domain ?
> It looks like a random subset construction - which does not constitute a separate domain in the reviewers opinion.
> In this point the reviewer still disagrees. Putting this in the title is misleading. Cross-domain would be to train something on stomach cancers and try it on colon cancer. Or train it on PCBs (printed circuit boards) and evaluate it on something different from a PCB.
>
> **[W3]** yes, have seen that.
>
> _They put the central claim in the title of being robust to outliers in the training data. However there is only one measurement of performance for different contamination noise levels - for VisA and for a very limited range._
>
> - The reviewer stands with its opinion that it is only for VisA and for a very limited range.
> - That central point deserved more dataset and a more wide range of contaminations.
>     - In this point the reviewer still disagrees.
>
> **[W4]** the reviewer is aware of both. Again, the text in lines 121-122 point with a **:** to equation (1) which is not a distance. If so, then eq(1) should become eq (3) by a cyclic rotation, which would resolve that.
>
> **[W5]** ok
>
> **[W6]** it is correlated to [W1] ,
>
> **[W7,W8]** okay
>
> **[W9]** not okay. Then one needs to do some cross-validation or the like.  When other papers do that, it makes it not acceptable.  Test set hyperparameter optimization is not generalizable.
>
> For [W1] and by extension to [W6] , and for W4 the reviewer accepts the answer of the authors. For the other points the reviewer felt not convinced.
> The reviewer will increase the score a bit away from the very negative 2 but given the other points the reviewer still leans towards a reject level rating.

---

> > ### Author Response · Authors · 2025-08-05
> >
> > Thank you for your time and for providing detailed comments and suggestions during the review process. We will take your feedback into careful consideration during the revision of our work.

---

### Official Review · Reviewer_NmKz · 2025-06-24

**Clarity:** 4
**Significance:** 3
**Originality:** 4
**Rating:** 4
**Confidence:** 5

**Summary:**

This paper aims to tackle fully unsupervised anomaly detection (with contamination data). This paper pioneer adapting the conventional knowledge distillation (KD) paradigm to the FUAD task. However, due to the presence of anomalies in the training data, traditional KD methods risk learning anomaly representations well during training. To address this, This paper proposes a novel cross-domain distillation framework. First, by domain-specific training on data of lower anomaly ratio can mitigate the student’s tendency to overfit to teacher’s anomaly features. Then, the domain-specific student can produce pseudo-normal features when applied to other domains, cross-domain knowledge aggregation is performed to guide a global student to learn generalized normal representations across all samples.

**Questions:**

Please see Strengths And Weaknesses.

**Ethical Concerns:**

["NO or VERY MINOR ethics concerns only"]

**Final Justification:**

The author's responses have resolved some of my concerns. With comprehensively considering the concerns of other reviewers, I decide to maintain my rating as borderline accept.

**Limitations:**

yes

**Quality:**

3

**Strengths And Weaknesses:**

**Paper Strengths:**

1. The proposed cross-domain distillation framework is novel. I think the idea that domain partitioning and then knowledge aggregation is very interesting and indeed helps to weaken the influence of abnormal samples in training data.

2. The motivation behind the framework is also clear, domain partitioning and cross-domain knowledge distillation align well with these two assumptions.

3. Compared to previous FUAD methods (generate pseudo labels for samples and then train on pseudo-labeled normal samples, which relies heavily on whether the pseudo labels are correct). The method proposed in this article is more impressive, which iteratively weakens the influence of abnormal samples during the training process.

4. The idea of dividing domains and aggregating knowledge is not limited to the knowledge distillation based AD methods, this core idea is general and can be extended to other UAD paradigms.

**Paper Weaknesses:**

**[Major]**

1. Although these two assumptions are not hard to understand, a good method cannot be based solely on the assumptions, and there is a lack of corresponding evidence in the paper to support these two assumptions. If these two assumptions do not hold in real cases, then the proposed framework is not sufficiently reasonable. Therefore, relevant quantitative results or representative examples should be provided for these assumptions to increase their persuasiveness.

2. In line 206, “we use the features output by the global student of the previous epoch as the basis for confidence evaluation”. However, what about the first epoch? For the first epoch, how to obtain the confidence score? This is not clear.

3. The experimental evaluation is not sufficient. This paper only conducts experiments on MVTecAD and VisA datasets with noise ratio 0.1 and 0.05. The authors should conduct more experiments on other datasets (e.g., MVTecAD3D, MVTecAD-LOCO, BTAD, etc) to more comprehensively evaluate the method. And this paper should also conduct experiments with larger noise ratio (e.g., 0.2, 0.3, 0.4) to evaluate the robustness of the method when encountering severe contamination. Additionally, why does VisA use 0.05 noise ratio not the 0.1 ratio used in MVTecAD?

4. In figure 5, the qualitative results are too few, you should add more qualitative results in figure 5.

**[Minor]**

In the caption of figure 1, “ross-Domain” should be “Cross-Domain”, replacing “MVTec AD-noise-0.1” with “MVTecAD-noise-0.1” is better. In line 237, “. providing” should be “, providing”. In line 286, “MVTec AD-noise-{0.2,0.15}” should be “MVTec AD-noise-{0.02,0.15}” as in figure 3.

I think the idea in this paper is novel and interesting, and will have good inspiration for future FUAD works. It may also become a classic framework in FUAD research. However, due to the above weaknesses, I would like to give a borderline accept. If the author can address these issues well, I am glad to raise the score.

---

> ### Author Rebuttal · Authors · 2025-07-31
>
> Thank you for recognizing the novelty of our paper for FUAD task. Your feedback is crucial for improving our work, and we welcome any further questions.
>
> **[W1] Evidence supporting Two Key Assumptions**
>
> Thank you for emphasizing the importance of validating the two key assumptions of our framework.
>
> For the **first assumption**, the **theoretical** basis has already been provided in `Sec. 3.2`, where Eq. (7) shows that anomaly types with low occurrence probabilities have negligible influence during optimization, leading the student to learn normal features instead. This provides an intuitive and solid foundation. Additionally, we follow your suggestion and design a toy experiment to quantitatively verify this assumption (please refer to our responses to `Reviewer Gahk [W2]`).
>
> For the **second assumption**, supporting evidence has been provided in `Sec. A.2 of the supplementary material` through visualization experiments (`Figures A2–A4)`, demonstrating that domain-specific students produce correct anomaly maps on out-of-domain anomalies.
>
> We will make these points clearer and highlight the corresponding evidence in the revised manuscript.
>
>
>
> **[W2] About the calculation of the confidence score during the first epoch**
>
> Thank you for your insightful question.
>
> We acknowledge that our explanation in the paper is not sufficiently clear. For the first epoch ($e=0$), as shown in Eq. (15), $r(e) = r(0) = 0$, which means the **high-confidence set is empty** during the initial epoch. Therefore, in this case, it is unnecessary to compute $\text{Conf}_i$ in the first epoch.
>
> We will clarify this point in the revised version to avoid confusion for readers.
>
>
>
> **[W3] Insufficient experimental evaluation**
>
> Thank you for your valuable comment. Let's address your concerns point by point below:
>
> > **1. The paper only conducts experiments on MVTecAD and VisA datasets with noise ratios of 0.1 and 0.05.**
>
> We have compared RD and our proposed CDD method on the MVTec AD dataset with noise ratios ranging from 0.02 to 0.15, as depicted in `Fig. 3` of the original manuscript. Additionally, in the supplementary material, we have included `Fig. A5` to compare RD and CDD on the VisA dataset with noise ratios of 0.03, 0.05, and 0.07.
>
>
>
> > **2. The authors should conduct more experiments on other datasets (e.g., MVTecAD3D, MVTecAD-LOCO, BTAD, etc.) to more comprehensively evaluate the method.**
>
> We appreciate your suggestion and have included additional experiments on the MVTec3d (considering only RGB images) and BTAD datasets.
>
> For the **MVTec3d** dataset, we evaluate anomaly localization performance (P-AUC/PRO) with noise ratio=0.1, as shown in the tables below. Obviously, CDD achieves better performance under both *No Overlap* and *Overlap* settings compared to RD and SoftPatch.
>
> **(1) MVTec3d-noise-0.1: *No Overlap* Setting**
>
> No overlap
>
> |             | SoftPatch   | RD          | CDD (Ours)          |
> | ----------- | ----------- | ----------- | ------------------- |
> | bagel       | 0.965/0.946 | 0.985/0.941 | 0.989/0.960         |
> | cable gland | 0.983/0.988 | 0.993/0.990 | 0.988/0.988         |
> | carrot      | 0.992/0.977 | 0.995/0.986 | 0.994/0.987         |
> | cookie      | 0.963/0.907 | 0.966/0.851 | 0.972/0.861         |
> | dowel       | 0.998/0.988 | 0.995/0.990 | 0.993/0.985         |
> | foam        | 0.999/0.780 | 0.941/0.812 | 0.949/0.845         |
> | peach       | 0.983/0.940 | 0.991/0.967 | 0.993/0.971         |
> | potato      | 0.952/0.948 | 0.991/0.979 | 0.991/0.976         |
> | rope        | 0.990/0.963 | 0.993/0.963 | 0.994/0.963         |
> | tire        | 0.994/0.975 | 0.995/0.989 | 0.994/0.981         |
> | Average     | 0.982/0.941 | 0.985/0.947 | **0.986**/**0.952** |
>
> **(2) MVTec3d-noise-0.1: *Overlap* Setting**
>
> |             | SoftPatch   | RD          | CDD (Ours)          |
> | ----------- | ----------- | ----------- | ------------------- |
> | bagel       | 0.887/0.895 | 0.844/0.927 | 0.987/0.952         |
> | cable gland | 0.912/0.955 | 0.910/0.988 | 0.974/0.989         |
> | carrot      | 0.936/0.923 | 0.931/0.985 | 0.992/0.987         |
> | cookie      | 0.893/0.816 | 0.836/0.816 | 0.949/0.848         |
> | dowel       | 0.998/0.973 | 0.881/0.990 | 0.984/0.987         |
> | foam        | 0.999/0.729 | 0.835/0.824 | 0.937/0.853         |
> | peach       | 0.941/0.853 | 0.867/0.951 | 0.970/0.967         |
> | potato      | 0.826/0.702 | 0.951/0.970 | 0.986/0.973         |
> | rope        | 0.993/0.936 | 0.619/0.916 | 0.986/0.948         |
> | tire        | 0.949/0.939 | 0.891/0.981 | 0.993/0.981         |
> | Average     | 0.933/0.872 | 0.856/0.935 | **0.976**/**0.948** |
>
> For the **BTAD** dataset, we test anomaly localization (P-AUC/PRO) with a noise ratio of 0.1 on classes '01' and '02' (as class '03' has insufficient samples to support a noise ratio of 0.1), with results shown below. We compare CDD with SoftPatch and RD, proving that CDD also has a good effect on the BTAD dataset.
>
> **(1) BTAD-noise-0.1: *No Overlap* Setting**
>
> | Category | SoftPatch   | RD          | CDD (Ours)          |
> | -------- | ----------- | ----------- | ------------------- |
> | 01       | 0.983/0.718 | 0.981/0.714 | 0.984/0.750         |
> | 02       | 0.942/0.509 | 0.969/0.571 | 0.961/0.607         |
> | Average  | 0.963/0.613 | 0.975/0.643 | **0.972**/**0.679** |
>
> **(2) BTAD-noise-0.1: *Overlap* Setting**
>
> |         | SoftPatch   | RD          | CDD (Ours)          |
> | ------- | ----------- | ----------- | ------------------- |
> | 01      | 0.679/0.538 | 0.911/0.717 | 0.961/0.736         |
> | 02      | 0.891/0.494 | 0.873/0.562 | 0.954/0.592         |
> | Average | 0.785/0.516 | 0.892/0.640 | **0.958**/**0.664** |
>
> > **3. The paper should conduct experiments with larger noise ratios (e.g., 0.2, 0.3, 0.4) to evaluate the robustness of the method under severe contamination.**
>
> In our experiments, a portion of anomalous samples from the test set was added to the training set to achieve the desired noise ratio. When the original training set has many normal samples and the test set has few anomalous samples, utilizing all normal samples limits the achievable noise ratio. Considering all categories in MVTec AD, we set the maximum noise ratio to 0.15.
>
> However, we agree that evaluating higher noise ratios is valuable for assessing robustness under severe contamination. Following your suggestion, we conduct anomaly localization (P-AUC/PRO) comparisons for noise ratios of 0.2, 0.3, and 0.4 for several main methods, with results shown below. (To achieve these noise ratios, we appropriately reduce the number of normal samples in the training set.)
>
> **(1) MVTec AD-noise-0.2/0.3/0.4: *No Overlap* Setting**
>
> | Noise_ratio | SoftPatch           | RD          | CDD (Ours)          |
> | ----------- | ------------------- | ----------- | ------------------- |
> | 0.2         | **0.970**/**0.916** | 0.956/0.908 | 0.969/0.911         |
> | 0.3         | 0.958/0.912         | 0.944/0.903 | **0.967**/**0.918** |
> | 0.4         | 0.954/0.912         | 0.945/0.906 | **0.966**/**0.913** |
>
> **(2) MVTec AD-noise-0.2/0.3/0.4: *Overlap* Setting**
>
> | Noise_ratio | SoftPatch   | RD          | CDD (Ours)          |
> | ----------- | ----------- | ----------- | ------------------- |
> | 0.2         | 0.941/0.901 | 0.811/0.890 | **0.957**/**0.909** |
> | 0.3         | 0.906/0.868 | 0.745/0.857 | **0.957**/**0.910** |
> | 0.4         | 0.890/0.842 | 0.746/0.856 | **0.944**/**0.902** |
>
> The above experimental results prove that CDD maintains a stable and superior anomaly localization effect compared to the baseline RD and SoftPatch under various noise ratios, especially under the *Overlap* setting. These results show that CDD can alleviate the interference of anomaly noise in the training set
>
>
>
> >  **4. Why does VisA use a 0.05 noise ratio instead of the 0.1 ratio used in MVTecAD?**
>
> The primary experiments on VisA used a noise ratio of 0.05 because some categories in the VisA dataset have significantly more normal samples than available anomalous samples. We will include additional experiments with higher noise ratios in the revised manuscript.
>
>
>
> **[W4] More qualitative results**
>
> Thank you for your suggestion.
>
> We totally agree with you. However, due to the page limit for the main submission, it was not feasible to include more qualitative examples in Fig. 5. To address this limitation, we have provided additional qualitative results in the supplementary material. Please refer to `Fig. A6, A8, A9, and A10 in the supplementary material` for more comprehensive visualizations.

---

> ### Comment · Reviewer_NmKz · 2025-08-01
>
> Thanks for your careful responses. To be honest, Eq.7 cannot be considered a theory at all, it is only about the gradient of loss, and the equations in Section 3.2 cannot be considered as theoretical proofs. Theory needs to be rigorously proven, and the equations in Section 3.2 only can be seen as formal explanations. Only relying on a toy experiment to validate the first assumption is not enough. This assumption is counter-intuitive: the more anomalies of a certain type during training, the worse the results during testing (more training samples can't result in better results). Your validation needs to be based on AD datasets and provide extensive results on many classes (e.g, results of 15 classes from MVTec). For the second assumption, it is also best to provide quantitative results under cross-domain conditions.
>
> For [W2], why "it is unnecessary to compute $\text{Conf}_i$ in the first epoch".

---

> > ### Author Response · Authors · 2025-08-03
> > **Response to Reviewer NmKz: Part I**
> >
> > Thank you for carefully reviewing our rebuttal and for raising these thoughtful questions. We appreciate the opportunity to address any remaining concerns. Below, we provide detailed responses to both of your questions.
> >
> > ### [Q1-new] Validation of Assumption 1 and Assumption 2
> >
> > **Regarding Assumption 1**
> >
> > First, we would like to clarify a potential misunderstanding caused by our previous rebuttal experiments. The original intent of Assumption 1 is that **reducing the probability of anomalies during training helps prevent the student from learning teacher representations of anomalies**.
> >
> > In the proposed CDD framework, this assumption is applied with the purpose of **reducing the sensitivity of domain-specific students to anomalies within their respective domains**. Our strategy achieves this by **injecting high-confidence samples, which are likely normal, into each domain as normal-like samples (forming overlaps among domains)**. In other words, we reduce anomaly probability by adding more normal samples, rather than by removing anomalies. Therefore, this design is not counter-intuitive.
> >
> > However, in the previous toy experiments, which differ from the actual application in CDD, we simulated the effect by **removing anomalous samples to reduce their probability**. The results still demonstrate that reducing anomaly probability improves detection performance. Similar findings can be observed in the experiments for `Reviewer Gahk [W1]`, where RD trained with noise ratios of 0.0 and 0.1 (using the same test set) shows consistent trends across 15 classes. This validates that reducing anomaly probability mitigates the student’s learning of anomalous teacher features, thereby improving anomaly detection.
> >
> > We partially understand your concern that in our toy experiments, reducing anomalies also removes some normal information within those samples, which might appear counter-intuitive given the performance improvement. However, the underlying reason is twofold: (1) Normal samples are highly similar in representation, so even a reduced number of normal samples can provide sufficient normal information; (2) More importantly, the removed anomalous samples contain noise that interferes with unsupervised anomaly detection, and eliminating this interference outweighs the minor loss of normal information, resulting in improved performance.
> >
> > To further address your concern and provide evidence more aligned with how Assumption 1 is used in our framework, we conduct additional experiments simulating the process of **gradually adding different amounts of high-confidence samples into each domain**, while keeping the number of anomalies (of the same anomaly type) fixed. All experiments use the same test set, which consists of normal samples and anomalous samples of the same anomaly type as the training anomalous samples. The PRO results across 15 classes are reported **Table 1** below. These results show that **as more normal samples are added (reducing anomaly probability), the model achieves better anomaly detection performance**, suggesting that the student does not overfit to the teacher’s anomalous features.
> >
> > > **Table1:  Quantitative PRO results of the verification experiments of Assumption 1**
> >
> > |Anomaly Samples||5||
> > |-|-|-|-|
> > |**Normal Sample**|**30**|**40**|**50**|
> > |bottle|0.978|0.978|0.981|
> > |cable|0.105|0.113|0.119|
> > |capsule|0.879|0.916|0.926|
> > |carpet|0.725|0.749|0.884|
> > |grid|0.988|0.989|0.989|
> > |hazelnut|0.927|0.931|0.940|
> > |leather|0.989|0.992|0.993|
> > |metal_nut |0.881|0.920|0.916|
> > |pill |0.964|0.967|0.969|
> > |screw|0.958|0.967|0.968|
> > |tile|0.467|0.644|0.784|
> > |toothbrush|0.509|0.513|0.496|
> > |transistor | 0.824  | 0.801|0.838|
> > |wood|0.778|0.861|0.921|
> > |zipper|0.945|0.949|0.972|
> > |Average|0.794|0.819|0.846|
> >
> > In practice, defining anomaly types is difficult because anomalies are highly diverse—even within a certain labeled anomaly type in the dataset, the actual anomalous regions can vary significantly. To make the evaluation more stringent, we modify the test set: for anomalies, only those that appear in the training set are retained. **Table 2** shows that after adding normal samples, even when the student has seen the same anomalous samples during training, it tends to learn more normal patterns, leading to improved detection results.
> >
> > > **Table 2:  Quantitative PRO results of the more strict verification experiments of Assumption 1**
> >
> > | Anomaly Samples ||5||
> > |-|-|-|-|
> > | **Normal Samples**|**30**|**40**|**50**|
> > |bottle|0.961|0.967|0.971|
> > |cable|0.002|0.003|0.005|
> > |capsule|0.794|0.928|0.947|
> > |carpet| 0.093| 0.176| 0.628|
> > |grid|0.993|0.995| 0.994|
> > |hazelnut|0.857|0.868|0.902|
> > |leather|0.983| 0.994  | 0.995  |
> > |metal_nut | 0.759  | 0.881  | 0.848  |
> > | pill | 0.959| 0.954| 0.965 |
> > | screw|0.983|0.983|0.958  |
> > | tile|0.299  | 0.583  | 0.863  |
> > | toothbrush | 0.449  | 0.440  | 0.416  |
> > | transistor| 0.827  | 0.780  | 0.840  |
> > | wood| 0.697  | 0.815  | 0.949  |
> > | zipper|0.919  | 0.923  | 0.957  |
> > | Average|0.705| 0.753 |0.816  |

---

> > ### Author Response · Authors · 2025-08-03
> > **Response to Reviewer NmKz: Part II**
> >
> > ### [Q1-new] Validation of Assumption 1 and Assumption 2
> >
> > **Regarding Assumption 2**
> >
> > Following your suggestion, we conduct a quantitative experiment to validate Assumption 2. We designed two training settings:
> >
> > - (1) 45 normal samples and 5 anomalous samples (all of the same anomaly type);
> > - (2) 50 normal samples.
> >
> > The test set remains the same in both cases, consisting of normal samples and anomalies of types not seen during training. The results are shown in **Table 3**. We can observe that the **anomaly detection performance on unseen anomaly types is similar under both settings**. This confirms Assumption 2: even if a student learns the teacher’s features for certain anomaly types during training, it still retains the ability to detect other anomaly types.
> >
> > > **Table 3:  Quantitative PRO results of the verification experiments of Assumption 2**
> >
> > | Settings   | (1) 45 normal + 5 anoamlous samples | (2) 50 normal samples |
> > | ---------- | ------------------------------- | ----------------- |
> > | bottle     | 0.952                           | 0.949             |
> > | cable      | 0.856                           | 0.865             |
> > | capsule    | 0.945                           | 0.939             |
> > | carpet     | 0.964                           | 0.963             |
> > | grid       | 0.972                           | 0.973             |
> > | hazelnut   | 0.937                           | 0.939             |
> > | leather    | 0.988                           | 0.987             |
> > | metal_nut  | 0.861                           | 0.863             |
> > | pill       | 0.936                           | 0.939             |
> > | screw      | 0.942                           | 0.936             |
> > | tile       | 0.826                           | 0.829             |
> > | toothbrush | 0.439                           | 0.433             |
> > | transistor | 0.757                           | 0.761             |
> > | wood       | 0.894                           | 0.895             |
> > | zipper     | 0.933                           | 0.948             |
> > | Average    | 0.880                           | 0.881             |
> >
> >
> >
> > ### [Q2-new] Confidence Computation in the First Epoch
> >
> > Thank you for the follow-up question about $\text{Conf}_i$ in the first epoch.
> >
> > To clarify, the confidence score in the first epoch does not affect the domain division process. In practice, we initialize a global student $\mathcal{S}^{Glo} _ {-1}$ before training begins. For consistency, we still compute $\text{Conf} _ i$ at $e=0$ using this initialized global student. However, since $\mathcal{S}^{Glo} _ {-1}$ is randomly initialized, the computed confidence scores are essentially random and carry no semantic meaning.
> >
> > Moreover, as shown in Eq. (13), $r(e)=r(0)=0$ for the first epoch, meaning the size of the high-confidence domain is $r(e)\cdot N = 0$. Therefore, **the first epoch always uses uniform domain division**, and the computed confidence values have no influence on the process.
> >
> > In short, while we compute $\text{Conf}_i$ for implementation consistency, it does not affect the training outcome in the first epoch. Confidence-guided strategies only start to play a role from the second epoch onward.
> >
> > We will make this clearer in the revised manuscript to avoid ambiguity.

---

> > > ### Comment · Reviewer_NmKz · 2025-08-04
> > >
> > > Thanks for your further responses, these have addressed my main concerns.

---

> > > > ### Author Response · Authors · 2025-08-05
> > > >
> > > > Thank you for your detailed follow-up and for acknowledging our additional clarifications—we appreciate your thorough engagement with our work.

---

### Official Review · Reviewer_Gahk · 2025-07-02

**Clarity:** 3
**Significance:** 2
**Originality:** 2
**Rating:** 3
**Confidence:** 5

**Summary:**

This paper studies fully unsupervised anomaly detection where training data include noises. A new algorithm is proposed which first divides the training set into multiple domains with lower anomaly ratios and train a domain-specific student for each. Then, pseudo-normal features are generated by domain-specific students collaboratively guide a global student to learn generalized normal representations across all samples. Experiments on several datasets verify the effectiveness of the proposed method.

**Questions:**

See the weakness above.

**Ethical Concerns:**

["NO or VERY MINOR ethics concerns only"]

**Limitations:**

Yes.

**Quality:**

3

**Strengths And Weaknesses:**

Strengths
- The proposed method is technically sound.
- Experiments on several datasets shows the advantage of the proposed method over existing methods.
- The paper is well presented.

Weakness
- There misses an important baseline. If the key motivation to study the fully unsupervised anomaly detection is to fully utilize the training data, an important baseline would be a method trained with noise-free data. Only by comparing a baseline method like this with demonstrated improvement, the benefit of training with noisy data can be justified. However, I did not see such a baseline employed for comparison.

- The two key assumptions lying the foundation of the proposed method have not been enough validated through rigorous proofs or specific experiments. While indeed there are some analytic reasoning for the first assumption, there is only some explanatory descriptions for the second assumption. From the perspective of the experiments, the final results indeed partially validate the soundness of the two assumptions, it is still necessary to design some experiments to validate each assumption separately.

- Which is the effect of further increasing the noise level? Will the proposed method show greater advantages over existing methods.

-  The proposed method employed a power teacher model, which gives the proposed method additional advantage over the existing methods.

- The improvements over the existing method are not significant.

- The performance difference of different design in the ablation study is very small, hard to validating the effectiveness.

---

> ### Author Rebuttal · Authors · 2025-07-31
>
> Thank you for your meticulous review of our experimental design, which provides critical insights for enhancing our paper. Please feel free to raise any further questions.
>
> **[W1] Missing Noise-Free Baseline Comparison**
>
> Thank you for your insightful comment about the need for a noise-free baseline to justify the benefits of training with noisy data in Fully Unsupervised Anomaly Detection (FUAD).
>
> First, we would like to clarify that the motivation for Fully Unsupervised Anomaly Detection (FUAD) is to **prevent the contamination of training data with noisy anomalous samples** (lnes 31–34). A natural idea in this context is to progressively filter out noisy data during training. However, in practice, since the filtering threshold is generally looser than the actual noise ratio, the retained samples are inevitably fewer than the true number of normal samples in the training set. Furthermore, even after filtering, the remaining training set may still contain anomalous noise samples. Therefore, this type of approach usually yields results **inferior to those obtained by training only on noise-free data** (which is equivalent to manually filtering anomalies).
>
> **Fully utilizing the training data is actually our motivation for domain partitioning and cross-domain training** (lines 196–199). We believe the filtering-based idea discussed above may limit performance, which is why we propose the cross-domain distillation method. In our approach, we do not remove anomalous noise samples; instead, these samples are leveraged during distillation training.
>
> We fully agree with your point: if our CDD can still outperform the baseline even when compared against a model trained on noise-free data, this would more clearly demonstrate that our method effectively utilizes the entire training set, including anomalous samples, to assist learning.
>
> Following your suggestion, we implement a new comparison experiment, as shown in the table below. For fairness, the same test set is used across all experiments. The second column of the table shows the performance of the baseline RD trained on noise-free data, which can be considered the upper bound of the previous idea. The third and fourth columns present RD and our proposed CDD trained on training data with noisy anomalous samples, respectively. As shown below, RD suffers from performance degradation under noisy conditions. In contrast, our method not only avoids this negative effect and remains close to the upper-bound result, but also improves the localization metric by 0.1%.
>
> | Noise_ratio | 0.0 (upper bound) | 0.1  | 0.1        |
> | ----------- | ----------------- | ---- | ---------- |
> | Method      | RD                | RD   | CDD (Ours) |
> | I-AUC (%)   | 98.5              | 97.2 | 98.4       |
> | P-AUC (%)   | 98.0              | 95.7 | **98.1**   |
> | PRO (%)     | 92.9              | 91.6 | **93.0**   |
>
>
>
> **[W2] Insufficient Validation of Two Key Assumptions**
>
> We appreciate your insightful comments regarding the validation of the two key assumptions underlying our proposed method.
>
> For the **first assumption**, which states that *when a particular anomaly type is sufficiently rare in the training data, the student fails to learn its corresponding teacher anomaly features and instead tends to produce features that closely resemble normal patterns*, it is theoretically supported in `Sec. 3.2`. Specifically, Eq. (7) shows that the gradient contribution from each anomaly type is weighted by its occurrence probability, which means that when this probability is very small, its influence on the optimization process becomes negligible. Consequently, the student focuses on learning normal features rather than anomalous ones. This analytic reasoning provides a solid theoretical foundation for the assumption.
>
> In addition to this theoretical reasoning, we further include a **toy experiment** following your suggestion. The experiments are conducted on the *carpet* category of the MVTec AD dataset. We select the anomaly type 'color' and construct three training sets by combining 20 normal samples with 2, 5, or 10 anomaly samples of this 'color' type. We then train RD on each set and evaluate it on 'color' anomalies. As shown in the table below, fewer anomaly samples result in better anomaly detection performance (higher P-AUC and P-AP), which aligns with our assumption.
>
> | Anomaly samples | P-AUC | P-AP  |
> | --------------- | ----- | ----- |
> | 2               | 0.685 | 0.050 |
> | 5               | 0.660 | 0.049 |
> | 10              | 0.657 | 0.044 |
>
> For the **second assumption**, which states that *domain-specific students trained on different domains produce pseudo-normal features when applied to other domains*, we have provided clear evidence in `Sec. A.2 of the supplementary material`. Specifically, Figures A2–A4 present visualization experiments conducted on the three classes of the MVTec AD dataset. For example, Figure A4 shows that when the domain-specific student of domain B encounters in-domain anomalies such as 'misplaced', the generated anomaly maps are wrong. In contrast, when applied to out-of-domain anomalies such as 'cut _ lead', it generates accurate anomaly maps, demonstrating that its features for unseen anomalies resemble normal patterns.
>
> We will make these points more explicit and strengthen the discussion of both assumptions in the revised version of the manuscript.
>
>
> **[W3] Effect of Higher Noise Levels**
>
> Thank you for your insightful question.
>
> We conduct additional experiments on the MVTec AD dataset with higher noise ratios of 0.2, 0.3, and 0.4. The results (P-AUC/PRO) are as follows:
>
> **(1) MVTec AD-noise-0.2/0.3/0.4: *No Overlap* Setting**
>
> | Noise_ratio | SoftPatch           | RD          | CDD (Ours)          |
> | ----------- | ------------------- | ----------- | ------------------- |
> | 0.2         | **0.970**/**0.916** | 0.956/0.908 | 0.969/0.911         |
> | 0.3         | 0.958/0.912         | 0.944/0.903 | **0.967**/**0.918** |
> | 0.4         | 0.954/0.912         | 0.945/0.906 | **0.966**/**0.913** |
>
> **(2) MVTec AD-noise-0.2/0.3/0.4: *Overlap* Setting**
>
> | Noise_ratio | SoftPatch   | RD          | CDD (Ours)          |
> | ------- | -------- | ------ | ------- |
> | 0.2         | 0.941/0.901 | 0.811/0.890 | **0.957**/**0.909** |
> | 0.3         | 0.906/0.868 | 0.745/0.857 | **0.957**/**0.910** |
> | 0.4         | 0.890/0.842 | 0.746/0.856 | **0.944**/**0.902** |
>
> As the noise ratio increases, our CDD method consistently outperforms the baseline RD, demonstrating superior stability. Notably, under the *Overlap* setting, the performance advantage of CDD becomes more pronounced at higher noise ratios, highlighting its robustness under severe noise contamination. These findings will be included in the revised manuscript.
>
>
>
> **[W4] Power teacher model**
>
> Thank you for raising this point.
>
> We are somewhat puzzled by this concern and would like to clarify that our Cross-Domain Distillation (CDD) framework does not employ a particularly powerful teacher model. The teacher model used in CDD is consistent with that of the baseline RD, which is WideResNet-50, as described in lines 174-176.
>
> If the term "teacher model" refers to the baseline model, we note that CDD is based on RD, while SoftPatch is based on PatchCore. These are contemporaneous works, and we did not use a clearly superior baseline.
>
>
>
> **[W5] Limited Performance Improvements**
>
> Thank you for your comment.
>
> We understand your concern about the degree of performance improvement. It is important to note that our method is built upon RD (which is better in anomaly localization), whereas existing methods such as SoftPatch are based on PatchCore (which is better in anomaly detection). Besides, our method and existing methods follow different optimization strategies and focus on different aspects, meaning that they are not in direct conflict and could potentially be combined. Our contribution lies in introducing a cross-domain training strategy tailored for FUAD scenarios, which is orthogonal to prior methods.
>
> Moreover, unlike PatchCore-based methods that rely on memory banks for feature retrieval, our method requires no such additional storage during inference. And our method also has **significantly faster inference speed** (see `Fig. 1(b)`). These advantages are particularly critical for real-time industrial applications.
>
> For more detailed performance comparisons of the baseline RD and PatchCore, please refer to our responses to `Reviewer TmMo [W1]`.
>
> **[W6] Small Performance Differences in Ablation Study**
>
> Thank you for your observation. We agree that the individual performance changes of some modules appear modest, but we believe that this is expected because **each component is designed to address a different aspect of the FUAD challenge** rather than produce large standalone gains.
>
> For example, Domain Construction (D.C.) and Pseudo-Normal aggregation (P.N.) introduce domain division and cross-domain training, which improves anomaly localization (e.g., P-AUC from 0.9566 to 0.9731). However, this process reduces the number of normal samples per domain, causing the student to slightly underfit normal representations. Since image-level anomaly detection relies on the maximum anomaly score of each image, such underfitting can occasionally cause higher scores on normal images, leading to a small drop in I-AUC.
>
> Subsequent modules including Confidence-Guided Domain Construction (Conf.G.) and Confidence-Distillation (Conf.D.) are designed to mitigate this underfitting by prioritizing high-confidence samples and refining global supervision. These strategies restore image-level accuracy and improve training stability, ultimately achieving the best overall performance.
>
> Thus, while individual effects may appear incremental, the cumulative gain across all metrics (**+1.15% on I-AUC, +2.52% on P-AUC, and +1.66% on PRO**) is significant. We will clarify these interactions in the revised manuscript.

---

> > ### Comment · Reviewer_Gahk · 2025-08-05
> >
> > Thanks for the response. After reading the comments from other reviews and the response, I still have concerns about the experimental design and the performance. So I would like to keep my rating.

---

> > > ### Author Response · Authors · 2025-08-06
> > >
> > > Thank you for your time and careful review. If there are still any specific points you would like us to address, we would be glad to discuss them further.

---

### Official Review · Reviewer_NYVh · 2025-07-02

**Clarity:** 3
**Significance:** 2
**Originality:** 2
**Rating:** 3
**Confidence:** 4

**Summary:**

This paper focuses on a practical and challenging task: fully unsupervised anomaly detection (FUAD), where the training set contains unlabeled anomaly samples. The authors point out that under this setting, traditional knowledge distillation (KD) methods run the risk of over-simulating anomalous features in the training set by the student network, leading to degradation of the detection performance. To solve this problem, the paper proposes a framework called Cross-Domain Distillation (CDD). The framework consists of two core stages: 1) Domain-Specific Training (DST), which divides the dataset into multiple domains with low anomaly ratios and trains a specialized student network for each domain to reduce the learning of anomaly patterns; 2) Cross-Domain Knowledge Aggregation (CDKA), which is a method to reduce the learning of anomaly patterns; and 3) Cross-Domain Knowledge Aggregation (CDKA), which is a method to reduce the learning of anomaly patterns. Aggregation (CDKA), which utilizes student networks not trained on a specific domain to generate “pseudo-normal” features for samples in that domain, and uses this to guide a domain-wide network of students to learn a generalized normal pattern representation. The experimental results show that this method achieve SOTA performance on the problem of FUAD, and ablation study demonstrates the effectiveness of proposed framework.

**Questions:**

1. Please provide the exact formulation of the final loss function for the global student and justify this specific design choice.
2. The feature fusion strategy in Eq. 11 employs a simple summation across layers. Please provide an ablation study or a stronger justification for why this unweighted summation is preferred over other potential fusion techniques, as different layers capture distinct feature semantics.
3. On page 6, the paper claims that the confidence-guided strategy ensures "The normal distribution in each domain remains more similar to the overall normal distribution." However, this crucial claim is not substantiated with either theoretical analysis or empirical evidence. Please provide a justification for this statement.
4. The framework's effectiveness relies on Assumption 2. However, this assumption might be fragile. In scenarios where different anomaly types across domains share subtle, underlying features (e.g., similar textural defects or color shifts), could the "pseudo-normal" features generated from other domains become contaminated, thereby misleading the global student? Please discuss the robustness of your method to such violations of this core assumption.
5. The Consensus-driven Selection strategy (Eq. 17 & 18) selects the pseudo-normal supervision signal most similar to the features from the previous epoch's global student. This creates a potential risk of confirmation bias, where the global student is primarily guided towards patterns it already models well. This could prevent it from learning novel aspects of the true normal distribution or correcting its past errors, potentially leading to convergence in a sub-optimal local minimum. How does your framework mitigate this risk of self-reinforcement?

**Ethical Concerns:**

["NO or VERY MINOR ethics concerns only"]

**Final Justification:**

Many recent baselines are missing, including the more recent unsupervised methods are missed, like the RD++, DesTseg etc. These unsupervised methods could be easily transformed to handle the contamination scenario, like by using these method to first remove the high-confidence contamination samples in the training dataset and then using the filtered dataset to train a new model, which could be served as very reasonable baselines. More seriously, the two core assumptions are too strong and lack sufficient justification. Indeed, due to the strong generalization capability of pre-trained CNN, unseen anomalies could be well reconstructed.

**Limitations:**

yes

**Paper Formatting Concerns:**

nil

**Quality:**

3

**Strengths And Weaknesses:**

Strengths.

1. The writing is clear and easy to follow.
2. The proposed cross-domain distillation idea is intuitive.
3. The experimental design is rigorous.

Weaknesses.

1. The improvement of performance is limited: The performance gains of the proposed CDD framework, particularly in key metrics like Image-level AUC on the MVTec AD dataset, are marginal when compared to existing state-of-the-art FUAD methods (e.g., SoftPatch). This limited improvement does not seem to fully justify the significant increase in model complexity, which involves multi-stage training with multiple student networks and dynamic parameter scheduling. The modest gains challenge the practical value of adopting such a complex architecture over simpler, yet competitive, alternatives.

2. The final loss function for the global student is ambiguous: The paper introduces two separate loss terms for training the global student: Eq. 20 for cross-domain pseudo-normal feature distillation and Eq. 21 for distillation on high-confidence samples. However, the manuscript fails to specify how these two loss components are combined into a single, final objective function for optimization. It is unclear whether they are simply summed or combined with weighting coefficients. This omission is a critical flaw as it hinders the reproducibility of the work, which is a cornerstone of scientific research.

3. Justification for the feature layer fusion strategy is lacking: While the paper specifies the method for fusing layer-wise anomaly maps in Eq. 11, it provides no justification for this design choice. Different network layers capture features at varying semantic levels, and it is not self-evident that a simple, unweighted sum is the optimal fusion strategy. The authors do not discuss or ablate alternative approaches, such as weighted summation or taking the maximum value, which leaves a gap in the methodological rigor and the exploration of a key design component.

---

> ### Author Rebuttal · Authors · 2025-07-31
>
> Thank you for your insightful feedback and interesting questions, which offers valuable inspiration for improving our paper and the future work. Please feel free to raise any further questions.
>
> **[W1] Limited improvement of performance**
>
> Thank you for your insightful comment.
>
> We understand your concern about the modest gains in Image-level AUC on MVTec AD dataset. This is because SoftPatch and CDD are based on different baselines: SoftPatch relies on **PatchCore**, which has an advantage in image-level anomaly detection metrics, while CDD is built on **RD**. Therefore, we consider it acceptable that CDD is less competitive in I-AUC compared to SoftPatch. Please refer to our responses to `Reviewer TmMo [W1]` for a detailed comparison of PatchCore and RD.
>
> Benefiting from RD’s excellent anomaly localization capability, CDD performs outstandingly in localization metrics like PRO, especially in the *Overlap* setting (`Table 2`). Additionally, on the larger VisA dataset, CDD shows clear advantages over SoftPatch, including in I-AUC (`Table 3`).
>
> Moreover, our parameter settings are robust, effective across multiple datasets. Notably, **CDD’s model complexity during inference does not increase**, maintaining excellent inference efficiency as shown in `Fig. 1 (b)`.
>
> In addition, CDD provides the following practical benefits:
>
> 1. **No Storage Overhead**: Unlike SoftPatch, which requires memory banks (e.g., 2.9GB for VisA, and scaling with dataset size), CDD relies solely on model parameters.
> 2. **Faster Inference**: CDD avoids feature similarity comparisons with memory banks, resulting in superior inference speed (`Fig. 1 (b)`).
> 3. **Superior Localization**: Leveraging RD, CDD excels in anomaly localization.
> 4. **Novel FUAD Solution**: CDD introduces a new solution adaptable to various baselines (e.g., reconstruction or knowledge distillation), paving the way for future work.
>
> In summary, while CDD’s improvement in I-AUC is moderate, its **robust localization performance, inference efficiency, and storage advantages** make it a practical and effective solution for fully unsupervised anomaly detection.
>
>
>
> **[W2&Q1] About the loss function of the global student**
>
> Thank you for your valuable comment regarding the loss function of the global student.
>
> There may be a misunderstanding: the loss terms in Eq. (20) (cross-domain pseudo-normal feature distillation) and Eq. (21) (distillation on high-confidence samples) are not combined for optimization.
>
> Instead, they are used in separate training stages, with **each loss computed and backpropagated independently**, as shown in `Fig. 2`. We will revise the manuscript, particularly in Sec. 4.2, to clearly indicate that these are two distinct training stages for better clarity.
>
>
>
> **[W3&Q2] About the design of anomaly map fusion strategy**
>
> Thank you for your valuable comment regarding the design of our anomaly map fusion strategy in Eq. (11) .
>
> We acknowledge your observation that the simple summation approach may appear straightforward. However, the primary focus of this paper is to propose a viable training framework for FUAD through Cross-Domain Distillation (CDD), rather than optimizing the anomaly map fusion strategy.
>
> **Our current anomaly map fusion strategy design intentionally aligns with the baseline RD** to ensure a fair comparison and demonstrate CDD’s adaptability to FUAD tasks. Optimizing the fusion strategy to adapt to specific data distributions could indeed improve performance, but it would obscure whether gains stem from our cross-domain distillation framework or the fusion strategy itself.
>
> We agree that exploring advanced fusion methods, such as weighted summation, is a promising direction and will include this in the discussion section.
>
>
>
> **[Q3] Clarification of the Claim on Confidence-Guided Strategy**
>
> Thank you for your insightful comment. We agree that the claim that  "the normal distribution in each domain remains more similar to the overall normal distribution" may have caused misunderstanding, as it might suggest a formal statistical notion of distribution, which was not our intention.
>
> Our intent is to convey a simpler idea: the confidence-guided strategy allocates samples with higher confidence scores (which are more likely to be normal) into each domain. As a result, each domain contains a larger portion of normal samples, making its normal set closer to the overall set of normal samples. This design ensures that domain-specific students better learn the teacher’s normal feature representations, which is critical for effective knowledge distillation.
>
> We will revise the wording in Section 4.1 to avoid the term "distribution" and instead clarify that our strategy aims to make **"each domain’s normal sample set closer to the overall normal sample set"**.
>
>
>
> **[Q4] Mitigating the Risk of Pseudo-Normal Feature Contamination Across Domains**
>
> Thank you for raising this important question. We fully agree that if anomalies across different domains share subtle, underlying similarities (e.g., similar textures or color shifts), Assumption 2 could be challenged, as some domain-specific students might inadvertently learn these features, potentially contaminating the pseudo-normal representations.
>
> We have actually considered this issue during the design of our framework and introduced multiple mechanisms to mitigate this risk:
>
> 1. **Domain-Specific Distillation with Regularization (Sec. 4.1)**
>
>    As described in lines 221–228, we introduce **a regularization term** when distilling each domain-specific student, leveraging the global student from the previous epoch as an auxiliary reference. This encourages domain-specific students to align with the global student’s representation, which is less likely to overfit to in-domain anomalies. In other words, this regularization constrains each domain-specific student to produce pseudo-normal features rather than only replicating teacher representations of anomalies within its own domain, thereby reducing contamination risk.
>
> 2. **Consensus-Driven Pseudo-Normal Feature Selection (Sec. 4.2)**
>
>    To further address this issue, we propose a consensus-based feature selection strategy that uses multiple domain-specific students to generate candidate features for a given sample, excluding the student from the sample’s own domain. Among these candidates, we select the feature most consistent with the previous global student’s representation for distillation. As stated explicitly in our manuscript, this design is intended to mitigate the impact of *"pseudo-normal feature contamination caused by some domain-specific students learning the ability to reconstruct certain types of teacher anomaly features (ines 238–239)."* This mechanism ensures that the features used to train the global student are robust even when anomaly similarities exist across domains.
>
> While these strategies have been discussed individually in the original text, we acknowledge that their role in addressing pseudo-normal feature contamination risk was not emphasized clearly enough. We will revise the manuscript to explicitly link these design components.
>
>
>
> **[Q5] Risk of Confirmation Bias in Consensus-Driven Selection Strategy**
>
> Thank you for highlighting this interesting concern.
>
> While we acknowledge this as a valid concern in theory, such confirmation bias is theoretically possible but practically negligible under our CDD framework. It requires very specific conditions:
>
> 1. **Initial Isolation**: Two highly similar anomalies (for example, $A_1, A_2$) must be assigned to different domains in the **first epoch**.  If both anomalies appear in the same domain initially, there will be no domain-specific students trained outside this domain that can reconstruct the teacher features corresponding to these anomalies, which means the global student can learn the pseudo-normal feature. In this way, bias will not occur from the beginning.
> 2. **Persistent Isolation**: In the remaining $E-1$ epochs, they must never appear in the same domain for more than $T$ consecutive epochs (otherwise the bias would be corrected). $T$ is the lower bound of the number of epochs for cross-domain training to correct bias.
> 3. **No Correction**: If they co-occur in the same domain for $T$ epochs, domain-specific students will learn proper normal features and the bias will diminish.
>
> Given dynamic domain re-division at every epoch in our framework, the accumulation of confirmation bias is extremely unlikely.
>
> Let $K$ be the number of domains, $E$ be the total epochs, and $T$ be the correction threshold. We approximate **the conservative upper bound of the probability that the bias occurs** as:
> $$
> P_{\text{bias}}^{\text{upper}} = \underbrace{\frac{K-1}{K} } _ { P _ 1} \times \underbrace{\left(1-\left(\frac{1}{K}\right)^T\right)^{E-T}}_{P_2}
> $$
>
> - $P_1$: Initial isolation probability.
> - $P_2$: The conservative upper bound of the probability that both $A_1$ and $A_2$ never occur in the same domain consecutively for $T$ epochs during the remaining $E-1$ epochs.
>
> For a typical setting ($K=3, E=200, T=3$):
>
>
> $$
> P_{\text{bias}}^{\text{upper}} \lesssim 1\\%
> $$
>
> While the theoretical probability is already extremely low,  we appreciate this question as it highlights an interesting edge case, which we plan to further investigate in future work.

---

> ### Comment · Reviewer_NYVh · 2025-08-05
>
> Thanks for your response. The performance still looks uncompetitive to me. The visual defect detection problem on MVTech dataset has been studied a lot, and a huge amount of methonds have been proposed in recent two years. In Table 1, many recent unsupervised methods are missed, like the RD++, DesTseg etc., more recent baselines should be included. Moreover, these methods could be easily transformed to handle the contamination scenario, like by using these method to first remove the high-confidence contamination samples in the training dataset and then using the filtered dataset to train a new model, which could be served as very reasonable baselines.
> More seriously, the response does not disperse my concerns on the two most core assumptions on the method. The two assumptions are too strong and lack sufficient justification. Actually, due to the strong generalization capability of pre-trained CNN, unseen anomalies could be well reconstructed. As for the fusion strategy, its rationale behind is still lacked.

---

> > ### Author Response · Authors · 2025-08-08
> > **Response to Reviewer NYVh: Part I**
> >
> > Thank you for carefully reviewing our rebuttal. First, we'd like to apologize for the delay in responding, as we have been conducting the additional experiments you suggested. Below, we address your concerns point-by-point.
> >
> > ### [Q1-new] Comparison with other AD methods
> >
> > Our primary goal is to demonstrate the effectiveness of the training framework for Fully Unsupervised Anoamly Detection task, rather than to simply achieve small performance gains. Therefore, we consider comparison with RD that our method is based on as the most critical baseline.
> >
> > That said, our paper has already included comparisons with newer methods such as **ReContrast** in `Tables 1–3 of the main manuscript` and `Tables A3–A5 in the supplementary material`.
> >
> > In addition, following your suggestion, we have included results for RD++ (also based on RD) on MVTec AD-noise-0.1. The results below show that our CDD method outperforms RD++ in both *No Overlap* and *Overlap* settings.
> >
> > > **(1) MVTec AD-noise-0.1: *No Overlap* Setting (I-AUC / P-AUC / PRO)**
> >
> > | Method     | RD                    | RD++                  | CDD (Ours)                        |
> > | ---------- | --------------------- | --------------------- | --------------------------------- |
> > | bottle     | 0.997 / 0.983 / 0.955 | 1.000 / 0.986 / 0.960 | 1.000 / 0.987 / 0.959             |
> > | cable      | 0.931 / 0.835 / 0.768 | 0.971 / 0.874 / 0.806 | 0.981 / 0.969 / 0.891             |
> > | capsule    | 0.939 / 0.980 / 0.956 | 0.949 / 0.980 / 0.954 | 0.942 / 0.984 / 0.950             |
> > | carpet     | 0.985 / 0.988 / 0.957 | 0.974 / 0.986 / 0.963 | 0.989 / 0.989 / 0.960             |
> > | grid       | 0.956 / 0.994 / 0.979 | 0.985 / 0.995 / 0.980 | 1.000 / 0.992 / 0.976             |
> > | hazelnut   | 1.000 / 0.992 / 0.936 | 1.000 / 0.993 / 0.945 | 1.000 / 0.993 / 0.945             |
> > | leather    | 1.000/ 0.995 / 0.988  | 1.000 / 0.995 / 0.989 | 1.000 / 0.991 / 0.971             |
> > | metal_nut  | 0.988 / 0.833 / 0.859 | 0.995 / 0.839 / 0.857 | 1.000 / 0.962 / 0.870             |
> > | pill       | 0.960 / 0.966 / 0.956 | 0.958 / 0.971 / 0.956 | 0.971 / 0.978 / 0.958             |
> > | screw      | 0.980 / 0.995 / 0.983 | 0.973 / 0.995 / 0.980 | 0.934 / 0.992 / 0.974             |
> > | tile       | 0.988 / 0.961 / 0.858 | 0.995 / 0.968 / 0.884 | 0.997 / 0.955 / 0.879             |
> > | toothbrush | 1.000 / 0.991 / 0.939 | 1.000 / 0.990 / 0.934 | 0.997 / 0.987 / 0.916             |
> > | transistor | 0.943 / 0.882 / 0.753 | 0.947 / 0.877 / 0.747 | 0.998 / 0.980 / 0.831             |
> > | wood       | 0.990 / 0.978 / 0.906 | 0.993 / 0.982 / 0.924 | 0.993 / 0.979 / 0.916             |
> > | zipper     | 0.924 / 0.976 / 0.941 | 0.932 / 0.979 / 0.943 | 0.958 / 0.980 / 0.950             |
> > | Average    | 0.972 / 0.957 / 0.916 | 0.978 / 0.961 / 0.921 | **0.984** / **0.981** / **0.930** |
> >
> > > **(2) MVTec AD-noise-0.1: *Overlap* Setting (I-AUC / P-AUC / PRO)**
> >
> > | Method     | RD                    | RD++                  | CDD (Ours)                        |
> > | ---------- | --------------------- | --------------------- | --------------------------------- |
> > | bottle     | 0.742 / 0.917 / 0.928 | 0.902 / 0.955 / 0.947 | 1.000 / 0.982 / 0.949             |
> > | cable      | 0.709 / 0.727 / 0.740 | 0.739 / 0.791 / 0.768 | 0.970 / 0.962 / 0.888             |
> > | capsule    | 0.788 / 0.889 / 0.953 | 0.868 / 0.937 / 0.949 | 0.939 / 0.978 / 0.940             |
> > | carpet     | 0.675 / 0.732 / 0.932 | 0.783 / 0.862 / 0.959 | 0.990 / 0.988 / 0.967             |
> > | grid       | 0.947 / 0.990 / 0.966 | 0.976 / 0.992 / 0.978 | 1.000 / 0.989 / 0.971             |
> > | hazelnut   | 0.443 / 0.774 / 0.930 | 0.443 / 0.823 / 0.942 | 0.986 / 0.983 / 0.936             |
> > | leather    | 0.762 / 0.848 / 0.988 | 0.975 / 0.975 / 0.990 | 1.000 / 0.990 / 0.970             |
> > | metal_nut  | 0.755 / 0.736 / 0.830 | 0.762 / 0.763 / 0.853 | 0.968 / 0.960 / 0.858             |
> > | pill       | 0.783 / 0.889 / 0.957 | 0.785 / 0.932 / 0.956 | 0.963 / 0.976 / 0.947             |
> > | screw      | 0.716 / 0.865 / 0.978 | 0.752 / 0.974 / 0.977 | 0.849 / 0.974 / 0.976             |
> > | tile       | 0.717 / 0.823 / 0.861 | 0.723 / 0.842 / 0.878 | 0.970 / 0.946 / 0.887             |
> > | toothbrush | 0.803 / 0.953 / 0.918 | 0.992 / 0.984 / 0.937 | 0.997 / 0.984 / 0.907             |
> > | transistor | 0.448 / 0.587 / 0.700 | 0.450 / 0.694 / 0.697 | 0.996 / 0.961 / 0.788             |
> > | wood       | 0.594 / 0.658 / 0.888 | 0.631 / 0.742 / 0.911 | 0.985 / 0.945 / 0.889             |
> > | zipper     | 0.745 / 0.888 / 0.942 | 0.891 / 0.964 / 0.942 | 0.957 / 0.971 / 0.938             |
> > | Average    | 0.708 / 0.818 / 0.901 | 0.778 / 0.882 / 0.912 | **0.971** / **0.973** / **0.921** |

---

> > ### Author Response · Authors · 2025-08-08
> > **Response to Reviewer NYVh: Part II**
> >
> > ### [Q2-new] About transforming other methods to handle contamination
> >
> > We understand your suggestion of adapting existing methods by filtering high-confidence contamination samples and retraining. However, this is your proposed variant rather than an established baseline, and we'd like to respectfully disagree that this is necessary, as it’s a hypothetical approach, not an established method.
> >
> > But we understand your concern and therefore conduct additional experiments to examine the potential of your suggested approach. The *upper bound* of this idea—training directly on all clean normal samples—represents an idealized condition that is not achievable in real-world FUAD settings. Our results (I-AUC / P-AUC / PRO) evaluated on the same test set in the table below show that CDD, even when trained with noisy data, reaches performance comparable to this upper bound of RD. This suggests that CDD can effectively utilize even contaminated samples to enhance detection, whereas the proposed filtering approach would inherently lose this potential. We also show that our CDD using the baseline RD achieves performance comparable to the more advanced baseline RD++ trained on clean normal data.
> >
> > | Noise_ratio | 0.0 (upper bound)     | 0.0 (upper bound)     | 0.1                   |
> > | ----------- | --------------------- | --------------------- | --------------------- |
> > | Method      | RD++                  | **RD (baseline)**     | CDD (Ours)            |
> > | bottle      | 1.000 / 0.987 / 0.960 | 1.000 / 0.986 / 0.957 | 1.000 / 0.987 / 0.959 |
> > | cable       | 0.976 / 0.977 / 0.907 | 0.976 / 0.965 / 0.872 | 0.981 / 0.969 / 0.891 |
> > | capsule     | 0.955 / 0.984 / 0.954 | 0.949 / 0.985 / 0.954 | 0.942 / 0.984 / 0.950 |
> > | carpet      | 0.984 / 0.987 / 0.963 | 0.981 / 0.988 / 0.955 | 0.989 / 0.989 / 0.960 |
> > | grid        | 1.000 / 0.994 / 0.978 | 1.000 / 0.994 / 0.978 | 1.000 / 0.992 / 0.976 |
> > | hazelnut    | 1.000 / 0.993 / 0.948 | 1.000 / 0.992 / 0.943 | 1.000 / 0.993 / 0.945 |
> > | leather     | 1.000 / 0.993 / 0.988 | 1.000 / 0.994 / 0.988 | 1.000 / 0.991 / 0.971 |
> > | metal_nut   | 1.000 / 0.971 / 0.886 | 1.000 / 0.969 / 0.884 | 1.000 / 0.962 / 0.870 |
> > | pill        | 0.968 / 0.977 / 0.958 | 0.966 / 0.977 / 0.957 | 0.971 / 0.978 / 0.958 |
> > | screw       | 0.982 / 0.996 / 0.983 | 0.986 / 0.996 / 0.981 | 0.934 / 0.992 / 0.974 |
> > | tile        | 1.000 / 0.959 / 0.886 | 0.989 / 0.950 / 0.868 | 0.997 / 0.955 / 0.879 |
> > | toothbrush  | 1.000 / 0.991 / 0.933 | 1.000 / 0.990 / 0.931 | 0.997 / 0.987 / 0.916 |
> > | transistor  | 0.992 / 0.965 / 0.840 | 0.966 / 0.950 / 0.806 | 0.998 / 0.980 / 0.831 |
> > | wood        | 1.000 / 0.977 / 0.909 | 0.997 / 0.975 / 0.901 | 0.993 / 0.979 / 0.916 |
> > | zipper      | 0.969 / 0.987 / 0.958 | 0.968 / 0.986 / 0.958 | 0.958 / 0.980 / 0.950 |
> > | Average     | 0.988 / 0.983 / 0.937 | 0.985 / 0.980 / 0.929 | 0.984 / 0.981 /0.930  |
> >
> > ### [Q3-new] About the two core assumptions
> >
> > It seems we may not have fully realized that you were expecting a more formal justification of the two assumptions, so this point was not sufficiently emphasized in our previous response.
> >
> > In fact, we have already provided extensive experimental validation of both assumptions in our responses to other reviewers, especially in the `follow-up response to Reviewer NmKz [Q1-new]`. If you are interested in the detailed experiments, please refer to that discussion (`Response to Reviewer NmKz: Part I`).
> >
> > We acknowledge that our wording may have made the assumptions appear stronger than intended. For example, Assumption 2 does not claim that unseen anomalies can *never* be reconstructed; rather, it states that our framework will not exacerbate overfitting to anomaly features compared to RD, especially in FUAD settings where fitting anomalous features is more likely. This nuance will be clarified in the revision.
> >
> > Furthermore, even in the presence of potential generalization, as you noted in Q4 of the original review regarding pseudo-normal feature contamination, our framework incorporates multiple mitigation mechanisms, as already detailed in our `initial response to you [Q4]`.
> >
> > ### [Q4-new] About the fusion strategy
> >
> > As we have stated in our `initial response to you [W3&Q2]`, our focus is not on marginal improvements from tuning tricks, but on investigating the FUAD framework itself. To ensure **fair comparison**, we intentionally retained RD’s original anomaly map fusion strategy rather than introducing additional modifications that could confound the evaluation of our framework.

---

### Official Review · Reviewer_TmMo · 2025-07-03

**Clarity:** 3
**Significance:** 2
**Originality:** 3
**Rating:** 4
**Confidence:** 4

**Summary:**

This paper is the first to explore the application of the knowledge distillation paradigm in fully unsupervised anomaly detection tasks. It addresses the over-generalization problem present in traditional knowledge distillation methods through two key strategies: Domain-Specific Training and Cross-Domain Knowledge Aggregation. Experimental results demonstrate its effectiveness.

**Questions:**

1.	In lines 34-37, it is mentioned that the 'Knowledge Distillation paradigm offers a storage-efficient alternative.' Could the authors provide relevant data on the storage overhead?
2.	In lines 56-57, the authors present their intuition. Could the authors justify the validity of this intuition from a theoretical or experimental perspective?
3.	Could evidence be provided to support the validity of the hypothesis presented in lines 62-63?

**Ethical Concerns:**

["NO or VERY MINOR ethics concerns only"]

**Final Justification:**

Thanks to the authors for their detailed response. The concerns I raised regarding the manuscript can be broadly categorized into two main areas: (1) concerns about the reported performance improvements, and (2) requests for the authors to substantiate certain viewpoints presented in the paper. Regarding the first category, the authors have provided some clarifications. As for the second, they have offered theoretical arguments or empirical evidence. In light of these responses, I assign a recommendation of Borderline Accept.

**Limitations:**

yes

**Quality:**

3

**Strengths And Weaknesses:**

**Strengths**

1.	This is the first work to explore the application of the knowledge distillation paradigm in fully unsupervised anomaly detection tasks.
2.	The presentation of the paper is relatively clear.
3.	The paper provides corresponding explanations for its assumptions.

**Weaknesses**

1.	The performance improvement compared to SoftPatch in Table 1 seems to be limited.
2.	The paper lacks evidence to support some of the viewpoints presented. For instance, the issue mentioned in lines 48-53, i.e., “This issue becomes more pronounced under the FUAD setting”, if supported by a simple experiment, I believe such evidence could further reinforce the motivation of the paper.

---

> ### Author Rebuttal · Authors · 2025-07-31
>
> Thank you for your thoughtful and constructive feedback on our paper. Your feedback is invaluable for improving our manuscript, and we will address each point thoroughly. Please feel free to raise any further questions.
>
> **[W1] Limited performance improvement over SoftPatch in Table 1**
>
> Thank you for your comment. We understand the concern regarding the seemingly modest performance improvement over SoftPatch in Table 1.
>
> It is important to note that this difference primarily arises from the **different underlying baselines and design focuses**: SoftPatch builds on **PatchCore**, which is optimized for **image-level anomaly detection**, while our proposed Cross-Domain Distillation (CDD) is developed on top of **RD**, which emphasizes **pixel-level anomaly localization**.
>
> To illustrate this inherent difference, the table below (source from the original papers) compares PatchCore and RD on the original MVTec AD dataset:
>
> | Metrics | PatchCore | RD    |
> | ------- | --------- | ----- |
> | I-AUC   | 0.991     | 0.985 |
> | P-AUC   | 0.981     | 0.978 |
> | PRO     | 0.934     | 0.939 |
>
> As shown above, PatchCore achieves a slightly higher I-AUC, whereas RD performs marginally better on PRO, reflecting their respective strengths. CDD's slightly lower I-AUC compared to SoftPatch is therefore expected, given RD’s localization-oriented design.
>
> Despite this, CDD achieves substantial improvements over RD under FUAD setting. Specifically, on MVTec AD-noise-0.1 (`Table 1 in the manuscript`), CDD improves RD by **1.2% in I-AUC**, **2.4% in P-AUC**, and **1.4% in PRO**.
>
> Beyond metric improvements, CDD offers two critical advantages for real-world deployment:
>
> 1. **Superior Localization**: CDD consistently outperforms SoftPatch in most localization metrics (P-AUC, PRO), as shown in `Tables 1–3`.
> 2. **Faster Inference**: Unlike PatchCore-based methods, CDD avoids memory bank operations, resulting in significantly faster inference (`see Fig. 1(b)`), which is crucial for industrial applications.
>
> Our primary goal is to adapt RD for the challenging FUAD setting, and the proposed strategy demonstrates clear benefits under this scenario. Future work will further explore boosting image-level metrics while maintaining these advantages.
>
>
>
> **[W2] Lack of evidence for the more pronounced over-generalization issue under FUAD setting (Lines 48–53)**
>
> Thank you for pointing this out.
>
> To clarify, in KD-based anomaly detection, the student network is trained to mimic the teacher’s feature representations on normal regions. At inference, anomalies are detected by measuring the feature discrepancy between teacher and student. Ideally, anomalies should yield large discrepancies because the student has never learned their patterns.
>
> However, an **over-generalization problem** may occur: even though the student is trained only on normal pixels, its learned feature representation ability may generalize to anomalies. This means **the student may also produce teacher-like features for anomalous regions**, reducing the teacher–student discrepancy that is critical for anomaly detection.
>
> The reason over-generalization issue becomes more severe under the FUAD setting is that **the training set contains anomalous samples**. This increases the chance that the student network learns teacher-like representations of anomalies during training, thereby reducing the feature discrepancy between teacher and student in anomalous regions, which is critical for anomaly detection.
>
> This phenomenon is theoretically supported by `Eq. (7) in Sec. 3.2`, where the student minimizes the feature difference with the teacher across all samples, including anomalous samples, leading to the student's overlearning of teachers' representation abilities on anomalous samples the under FUAD setting.
>
> To provide empirical evidence, we conduct a simple experiment on the MVTec AD dataset, comparing RD performance under two settings: (i) noise ratio = 0.0 (only use normal samples for training), and (ii) noise ratio = 0.1 (with 10% anomalous samples added into the train set). Both settings share the same  test set and normal samples used for training. The results are as follows:
>
> | Setting            | Noise Ratio | I-AUC | P-AUC | PRO   |
> | ------------------ | ----------- | ----- | ----- | ----- |
> | Unsupervised       | 0.0         | 0.985 | 0.980 | 0.929 |
> | Fully Unsupervised | 0.1         | 0.972 | 0.957 | 0.916 |
>
> The clear performance drop at noise ratio = 0.1, despite having more training data. It confirms that introducing anomalies into the training set causes the student to learn teacher-like anomaly patterns, reducing detection accuracy. This supports our claim that over-generalization becomes more pronounced under the FUAD setting, as described in lines 48–53.
>
>
>
> **[Q1] Storage overhead comparison (Lines 34-37)**
>
> Thank you for your insightful question.
>
> Lines 34-37 highlight that existing FUAD methods, such as SoftPatch [17], rely on **memory banks**, incurring significant storage overhead, while the methods based on Knowledge Distillation paradigm, such as our CDD, avoids this by storing only model parameters.
>
> The table below quantifies the **memory bank storage requirements for SoftPatch**:
>
> | Dataset  | Memory Bank Storage (GB) |
> | -------- | ------------------------ |
> | MVTec AD | 1.3                      |
> | VisA     | 2.9                      |
>
> These storage requirements **scale linearly with dataset size**, increasing storage demands for larger datasets. In contrast, KD paradigm eliminates memory banks, making it more storage-efficient.
>
> We will add this comparison to the revised manuscript to clarify the storage advantage of KD-based methods.
>
>
>
> **[Q2] Justification of the intuition on reducing anomaly probability during training (Lines 56-57)**
>
> We appreciate your request to justify the intuition in lines 56-57, which states that *"reducing the probability of anomalous samples being learned during training mitigates the student’s tendency to overfit to techer’s anomaly features"*. We wish to clarify that this intuition has been theoretically supported in `Sec. 3.2`, where we provide a detailed derivation to address this point.
>
> The intuition in lines 56-57 is actually a rephrasing of Assumption 1 (Sec. 3.2), which states that *when anomaly types are sufficiently rare in the training data, the student fails to learn the teacher’s anomaly features, instead producing features resembling normal patterns*. Both descriptions convey that a lower anomaly presence during training reduces the student’s learning anomalous teacher features, ensuring effective anomaly detection.
>
> Specifically, Eq. (7) in Sec. 3.2 models the gradient of the student’s parameters as
> $$
> \frac{\partial \mathcal{L}}{\partial \theta _ \mathcal{S}} = \mathbb{P}(\mathcal{N}) \cdot \mathbb{E} _ {I_i \sim \mathcal{N}}\left[\frac{\partial \ell _ {cos}}{\partial \theta _ \mathcal{S}}\right] + \sum _ {m=1}^{M_{train}} \mathbb{P}(\mathcal{A}_ m) \cdot \mathbb{E} _ { I _ j \sim \mathcal{A} _ m} \left[\frac{\partial \ell_{cos}}{\partial \theta_\mathcal{S}}\right]
> $$
> When the probability of anomalies $\mathbb{P}(\mathcal{A}m)$ is small, their contribution to the gradient is negligible, causing the student to prioritize learning normal features $\mathcal{N}$ over anomalous ones $\mathcal{A}_m$. This reduces the student’s tendency to mimic teacher anomaly features, as stated in lines 56-57. Thus, the theoretical derivation in Sec. 3.2 supports this conclusion, and we will clarify this connection in the revised manuscript.
>
>
>
> **[Q3] Evidence supporting the hypothesis that domain-specific students trained on different domains produce pseudo-normal features when applied to other domains (lines 62-63)**
>
> Thank you for this meaningful question regarding the hypothesis in lines 62-63, which states that *"domain-specific students trained on different domains produce pseudo-normal features when applied to other domains"*. We agree this is a critical point and have provided visualization evidence in `Sec. A.2 of the supplementary material`, specifically in **`Fig. A2, A3, and A4`**.
>
> For instance, Fig. A4 shows a toy visualization experiment conducted on the *transistor* class in MVTec AD dataset. The upper part of the figure shows that for in-domain anomalies, such as 'cut_lead', the student cannot generate accurate anomaly maps, while for out-of-domain anomalies, such as 'bent_lead', it produces anomaly maps well, suggesting that the student is able to generate pseudo-normal features for unseen anomalies. The lower part further proves this by simulating domain division, clearly showing that correct anomaly maps are generated for out-of-domain anomalies (unseen samples), which indicates domain-specific students generate the pseudo-normal features for these anomalies.
>
> These visualizations directly validate the hypothesis in lines 62-63, confirming that domain-specific students produce pseudo-normal features for unseen anomalies from other domain. We will clarify this connection in the revised manuscript.

---

> > ### Comment · Reviewer_TmMo · 2025-08-04
> >
> > Thank you for your detailed rebuttal. The previously unclear issues have been well addressed, and I will maintain my positive evaluation of this article.

---

> > > ### Author Response · Authors · 2025-08-05
> > >
> > > Thank you for your positive feedback. If there are any remaining concerns or suggestions, we would be happy to discuss them further.

---

### Decision · Program_Chairs · 2025-09-17

**Decision:**

Reject

**Comment:**

This work focuses on fully unsupervised anomaly detection (FUAD), where the training set contains unlabeled anomaly samples. The authors point out that under this setting, traditional knowledge distillation (KD) methods run the risk of over-simulating anomalous features in the training set by the student network, leading to degradation of the detection performance. To solve this problem, the paper proposes a framework called Cross-Domain Distillation (CDD). The framework consists of two core stages: 1) Domain-Specific Training (DST), which divides the dataset into multiple domains with low anomaly ratios and trains a specialized student network for each domain to reduce the learning of anomaly patterns; 2) Cross-Domain Knowledge Aggregation (CDKA), which is a method to reduce the learning of anomaly patterns; and 3) Cross-Domain Knowledge Aggregation (CDKA), which is a method to reduce the learning of anomaly patterns.

I think this is an interesting topic to address; however, in its current form, the paper can not be accepted. Too many open issues remain, as pointed out by the reviewers,  for example, concerns about the experimental design and the performance. I encourage the authors to address the issues addressed by the reviewers for a potential resubmission.